# A GIS-Based Approach for the Quantitative Assessment of Soil Quality and Sustainable Agriculture

**Mostafa A. Abdellatif** [1,2]**, Ahmed A. El Baroudy** [2]**, Muhammad Arshad** [3]**, Esawy K. Mahmoud** [2]**, Ahmed M. Saleh** [1]**, Farahat S. Moghanm** [4]**, Kamal H. Shaltout** [5]**, Ebrahem M. Eid** [6,7,*] **and Mohamed S. Shokr** [2,*]

1   National Authority for Remote Sensing and Space Science (NARSS), Cairo 11843, Egypt; mostafa.abdou@narss.sci.eg (M.A.A.); ahmedms@outlook.com (A.M.S.)
2   Soil and Water Department, Faculty of Agriculture, Tanta University, Tanta 31527, Egypt; drbaroudy@agr.tanta.edu.eg (A.A.E.B.); esawy.rezk@agr.tanta.edu.eg (E.K.M.)
3   Department of Chemical Engineering, College of Engineering, King Khalid University, Abha 61321, Saudi Arabia; moakhan@kku.edu.sa
4   Soil and Water Department, Faculty of Agriculture, Kafrelsheikh University, Kafr El-Sheikh 33516, Egypt; fsaadr@yahoo.ca
5   Botany Department, Faculty of Science, Tanta University, Tanta 31527, Egypt; kamal.shaltout@science.tanta.edu.eg
6   Biology Department, College of Science, King Khalid University, Abha 61321, Saudi Arabia
7   Botany Department, Faculty of Science, Kafrelsheikh University, Kafr El-Sheikh 33516, Egypt
*   Correspondence: ebrahem.eid@sci.kfs.edu.eg or eeid@kku.edu.sa (E.M.E.); mohamed_shokr@agr.tanta.edu.eg (M.S.S.)

**Abstract:** Assessing soil quality is considered one the most important indicators to ensure planned and sustainable use of agricultural lands according to their potential. The current study was carried out to develop a spatial model for the assessment of soil quality, based on four main quality indices, Fertility Index (FI), Physical Index (PI), Chemical Index (CI), and Geomorphologic Index (GI), as well as the Geographic Information System (GIS) and remote sensing data (RS). In addition to the GI, the Normalized Difference Vegetation Index (NDVI) parameter were added to assess soil quality in the study area (western part of Matrouh Governorate, Egypt) as accurately as possible. The study area suffers from a lack of awareness of agriculture practices, and it depends on seasonal rain for cultivation. Thus, it is very important to assess soil quality to deliver valuable data to decision makers and regional governments to find the best ways to improve soil quality and overcome the food security problem. We integrated a Digital Elevation Model (DEM) with Sentinel-2 satellite images to extract landform units of the study area. Forty-eight soil profiles were created to represent identified geomorphic units of the investigated area. We used the model builder function and a geostatistical approach based on ordinary kriging interpolation to map the soil quality index of the study area and categorize it into different classes. The soil quality (SQ) of the study area, classified into four classes (i.e., high quality (SQ2), moderate quality (SQ3), low quality (SQ4), and very low quality (SQ5)), occupied 0.90%, 21.87%, 22.22%, and 49.23% of the total study area, respectively. In addition, 5.74% of the study area was classified as uncultivated area as a reference. The developed soil quality model (DSQM) shows substantial agreement (0.67) with the weighted additive model, according to kappa coefficient statics, and significantly correlated with land capability $R^2$ (0.71). Hence, the model provides a full overview of SQ in the study area and can easily be implemented in similar environments to identify soil quality challenges and fight the negative factors that influence SQ, in addition to achieving environmental sustainability.

**Keywords:** developed soil quality model; GIS; Egypt; NDVI; geomorphologic index

## 1. Introduction

Globally, there are more than 800 million people who are chronically undernourished [1]. Africa has the highest proportion of people who suffer from chronic hunger [1].

Assessing and managing soil is considered one of the key ways to achieve food security by helping to bridge the food demand gap [2]. Political instability in most African countries affects agricultural practices and leads to underdeveloped and underexploited lands, having direct consequences for society [3].

In Egypt, agricultural lands are located in the Nile valley and the delta, which represent about 4% of the total area of Egypt [4]. The agricultural sector in Egypt plays a vital role in economic growth as it contributes 14.5%, 30%, and 41% of national gross domestic product, provision of foreign currency, and reducing unemployment, respectively [5].

The definition of soil quality (SQ) is the ability of soil included the ecosystem to supply plants with the nutrients needed throughout growth stages for the purpose of preserving crop yield [6–8]. Since SQ supports sustainable soil management as it is linked to soil productivity, a reliable assessment requires an accurate, multi-faceted quantification [9]. Maintaining soil productivity by soil quality management should be considered earnestly to ensure sufficient food for the burgeoning world population [10]. Soil quality is influenced by physical indicators such as bulk density, root depth, and soil texture, and chemical indicators, such as cation exchange capacity (CEC), electric conductivity (EC), and pH. There are highly significant correlations between these indictors and soil quality [11,12]. Soil quality could be negatively affected by conversion of land use as soil properties are significantly influenced by this practice, as it decreases, for example, soil organic carbon and total nitrogen. In addition, soil contamination by heavy metals may cause risks to humans and the ecosystem, decrease land suitability for agricultural production, and cause food insecurity and land tenure problems [13]. Potentially toxic elements have negative effects on plant growth, crop yield, and quality due to phytotoxicity [14]. Therefore, it is very important to focus on the sustainable use of agricultural lands to increase the soil quality [15,16]. To improve soil and water quality, precise measurements and efficient methods should be conducted [17]. Index indicators are the most appropriate method for assessing SQ [18]. Developing a soil quality index (SQI) requires selecting an indicator, scoring it, and then integrating scores into a single value [18]. The weighted additive index is one of the most used for SQ evaluation based on integrating indicator weights with corresponding scores [19]. The geometric mean algorism (GMA) and the nth root of a series of numbers are commonly utilized in assessments of desertification sensitivity and land suitability [20,21]. The GMA is used to characterize the data average or central tendency [22]. Analysis of land capability can be used to assess agricultural potential to face increasing drought impacts [23]. The land assessment concept belongs to the land performance rate and its capacity for crop production, while land capacity depends on many factors, such as location and the physical and chemical properties of soil, in addition to soil potential for agricultural production [24].

Currently, there are many land capability models established to introduce a quantified procedure to match land with actual and proposed uses, especially for arid and semi-arid regions, including the study area. An example is the Agricultural Land Evaluation System for arid and semi-arid regions (ALESarid), developed by Ismail et al. [25]. This model is combined with Geographic Information System (GIS) software to assess land capability and could provide a sensible solution given its accuracy, ease of application, and moderate data required [26]. GIS technology has enabled the spatial variability computation of different phenomena [27], including investigations of soil properties. Thus, combined GIS and geostatistical analyses can be very important in assessing the spatial variation of soil properties and those expected in un-sampled sites [28]. For instance, using variogram analyses can accurately map the complex spatial relationships between soil data layers [29]. One of the most commonly used interpolation methods is Kriging. This method is based on the identification of homogeneous subsets of similar yield-limiting factors; thus, it can sufficiently support precise farming [30,31].

The study area suffers from a lack of awareness of agriculture practices and scarcity of water as it depends on seasonal rain for cultivation [32]. As the study area is an important part of the existent economic activities in Egypt [33], the current study (1) identifies



geomorphologic units of the study area and (2) quantitatively assesses soil quality using the developed model based on four indicators, i.e., chemical, physical, fertility, and geomorphologic indices. In addition to the GI, the Normalized Difference Vegetation Index (NDVI) parameter were added as a new factors to reflect the specific soil quality and categorize it into different classes as accurately as possible. Finally, (3) the results from this model were validated with the weight additive index and correlated with land capability. To our knowledge, only few studies assess soil quality in the study area, so this paper offers valuable data to decision makers and regional governments to find the best ways to increase soil quality and overcome the food security problem, which is one of the most important challenges in the 2030 Agenda for Sustainable Development.

## 2. Materials and Methods

### 2.1. Study Area Description

The study area is located in the western part of Matrouh Governorate, Egypt. The studied area, Wadi Al-Halaazin located between longitudes 26°49′58.0′′ to 26°58′06′′ E and latitudes 31°13′07′′ to 31°26′36′′ N with total area 21369.74 ha (Figure 1). The international coastal road passes in the middle of it. Arid climate prevails in the study area as the average temperature reaches 18 °C in the winter and autumn but ranged from 18 °C to 25 °C in the summer. The range of rainfall is between 100 and 200 mm/year. In the winter and spring seasons, the vegetation cover changing due to rainfall is active and the natural vegetation spreads within study area, particularly in the wadis and streams in addition, the natural vegetation growth during the autumn and summer seasons because of the natural vegetation spreads on the fine sand stacks that keep rainwater [31] and barley is the main crop in the study area in addition small areas of some scattered olive trees (Figures 1 and 2). In the investigated area, the main geological units are Miocene and Quaternary deposits according to Ministry of Industry and Mineral Resources (MIMIR) [34] (Figure S1). The range of elevation is between 11 and 212 above sea level. The lowest elevation was noticed in the areas close to Mediterranean Sea, while the highest elevation located in the southern parts (Figure 2). [31]. The Normalized Difference Vegetation Index (NDVI) ranged from low (−0.22) to high (0.73) (Figure S2). The highest values of NDVI (expresses high intensity of cultivation) [35] were noticed in some small areas of study area (wadi unit) [31].

### 2.2. Extracting Landforms Units

The input data included: topographic data, information of spectral satellite (sentinel 2 acquired in April/2020), field surveys and stratigraphic characteristics. The NASA Shuttle Radar Topographic Mission (SRTM) Digital elevation model (DEM) (30 ∗ 30 resolution) (Figure 2) was chosen for study area. A simple filter by focal neighborhood statistics was used to decrease errors and noises. These noises occurred due to reclassification of topographic parameters. For the majority and mean focal statistic values, the largest and average values of the specified neighborhood pixels were assigned to the canter pixel of the moving window [36]. Algorithm of Planchon and Darboux [37] to correct DEM then corrected DEM was used to derive the indices of topographic (slope, aspect, plan curvature, profile curvature, slope length and steepness, relative slope position, valley depth, and analytical hill shading; Figure S3) with the SAGA GIS software [38]. By this method, we were able to identify the different landform units relied on satellite image visual interpretation and DEM in a 3D visualization mode, a hillshade in addition, field truth points [39] with the help of previous studies that were done on this area [40] trying to give the most appropriate nomenclature to landforms.

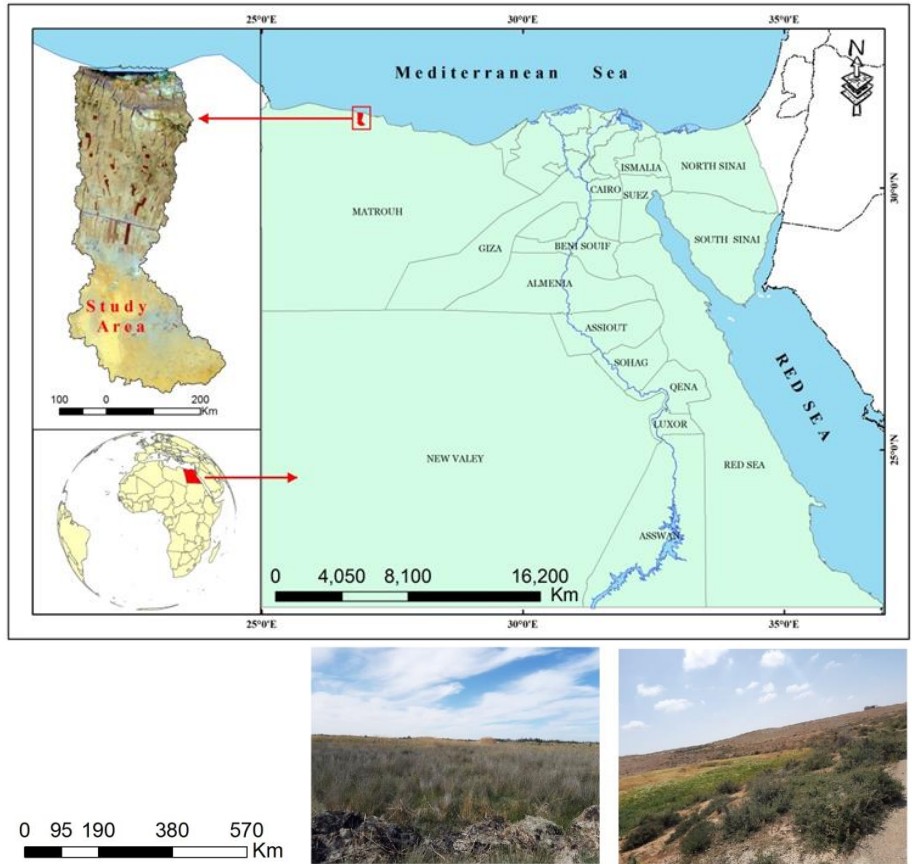

**Figure 1.** Location of investigated area.

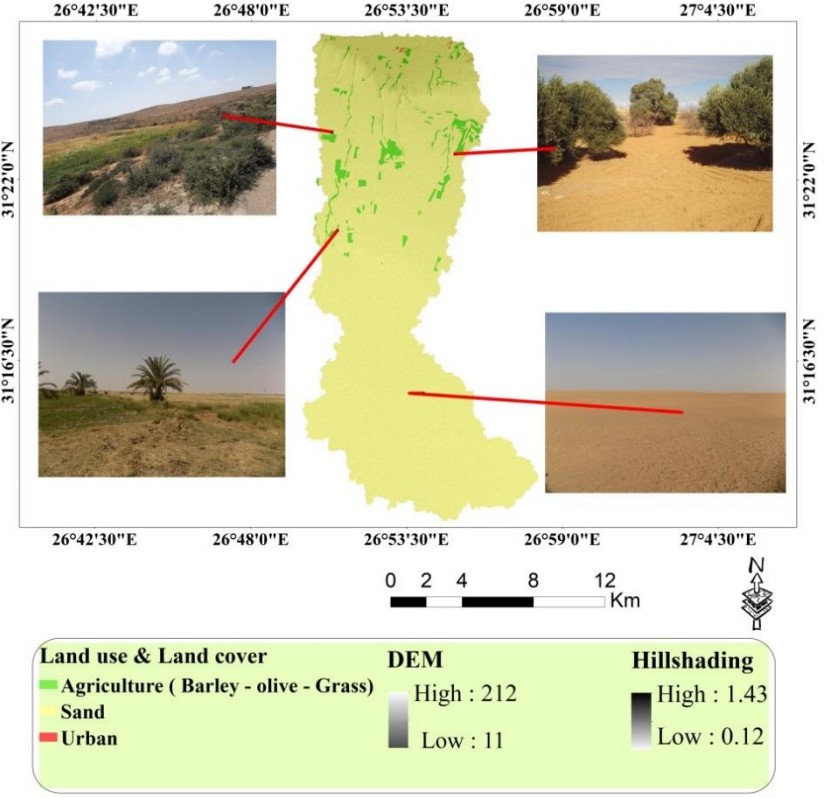

**Figure 2.** Land use and land cover of study area.

### 2.3. Calculation of NDVI

Calculation of NDVI was done on sentential 2 image (acquired in April 2020) of study area using raster calculator function in snap (V8) to subtract values of the Red (R) band from the Near-infrared (NIR) band, then divide by the sum of the R and NIR bands according to the following Equation (1):

$$\text{NDVI} = \frac{\text{NIR} - \text{IR}}{\text{NIR} + \text{IR}} \tag{1}$$

### 2.4. Collecting Samples and Laboratory Analyses

Forty-eight soil profiles were dug and distribution of them were depends on the identified geomorphic units of the investigated area (Figure 3). The depth of profiles is 150 cm or less relies on the hardpan presence. Soil profiles descriptions were done according to FAO [24]. Classification of soil profile was carried out according to USDA Soil Survey Staff [41]. The following chemical, physical, and biological properties of soil were determined: chemical analysis, i.e., salinity, soil reaction (pH), cation exchange capacity (CEC), and the exchangeable sodium percentage [42–46], physical properties, i.e., bulk density and the particle size distribution, hydraulic conductivity (HC) and water holding capacity (WHC) [47] and biological, i.e., soil organic matter content (OM%), available soil nitrogen (N), phosphorus (P), and potassium (K) [48,49]. Analysis was done in the accredited soil, water, and plant laboratory at Tanta University's Faculty of Agriculture in accordance with ISO/IEC 17,025:2017 requirements.

### 2.5. Distribution of Soil Properties

Ordinary Kriging (OK) is an advanced geostatistical procedure that can create a continuous surface from scattered soil samples depending on their characteristics [50]. In the present study, OK was chosen as the geostatistical model for estimating soil properties spatial distribution. Z(Xi) is supposed to be a regionalized variable with a variogram $\gamma(h)$, which is a function labelling the spatial aggregation field or random process Z(u). Methods of the exponential, Gaussian, spherical, and circular Equations (2)–(5) were used as the semi-variance model and choose of the best variogram based on the leave-one-out cross-validation results.

The exponential function was defined as the following:

$$Y(h) = \begin{cases} 0, h = 0 \\ C_0 + C\left(1 - e^{-\frac{h}{a}}\right), h > 0 \end{cases} \tag{2}$$

The Gaussian function was defined as:

$$Y(h) = \begin{cases} C_0 + C\left(1 - \exp\left(-\frac{h^2}{a^2}\right)\right), h > 0 \\ 0, h = 0 \end{cases} \tag{3}$$

The spherical function was defined as:

$$Y(h) = \begin{cases} C_0 + C\left(\frac{3h}{2a} - \frac{1}{2} - \left(\frac{h}{a}\right)^3\right), 0 < h \leq a \\ C_0 + C, \ h > a \\ 0, h = 0 \end{cases} \tag{4}$$

The circular function was defined as

$$Y(h) = \begin{cases} C_0 + C(1 - \frac{2}{\pi}\cos^{-1}\frac{h}{2} + \sqrt{1 - (h^2/a^2}), 0 < h \leq a \\ C_0 + C, \ h > a \\ 0, h = 0 \end{cases} \tag{5}$$

In these equations, a is the actual ranges for the spherical, circular, exponential, and Gaussian functions, respectively. h is the spatial lag, $C_0$ is the nugget, and C is the partial sill. The spatial variation of the soil samples for these variograms was isotropic. Traditional OK can introduce equitable estimates with minimum error. The OK function was expressed as:

$$Z(x_0) = \sum_{i=1}^{n} \lambda i(x_0) Z(x_i) \tag{6}$$

where: $\sum ni = 1\lambda i(x_0) = 1$; $Z \times (x_0)$ is the predicted value of variable z at location $x_0$; $Z(x_i)$ is the measured data; $\lambda i(x_0)$ refers to the weights linked with the measured values; and n is the number of predicted values within certain neighbor soil samples. The OK was applied utilizing the Create Fishnet tool in ArcGIS (Version 10.7, Esri, Inc., Redlands, CA, USA).

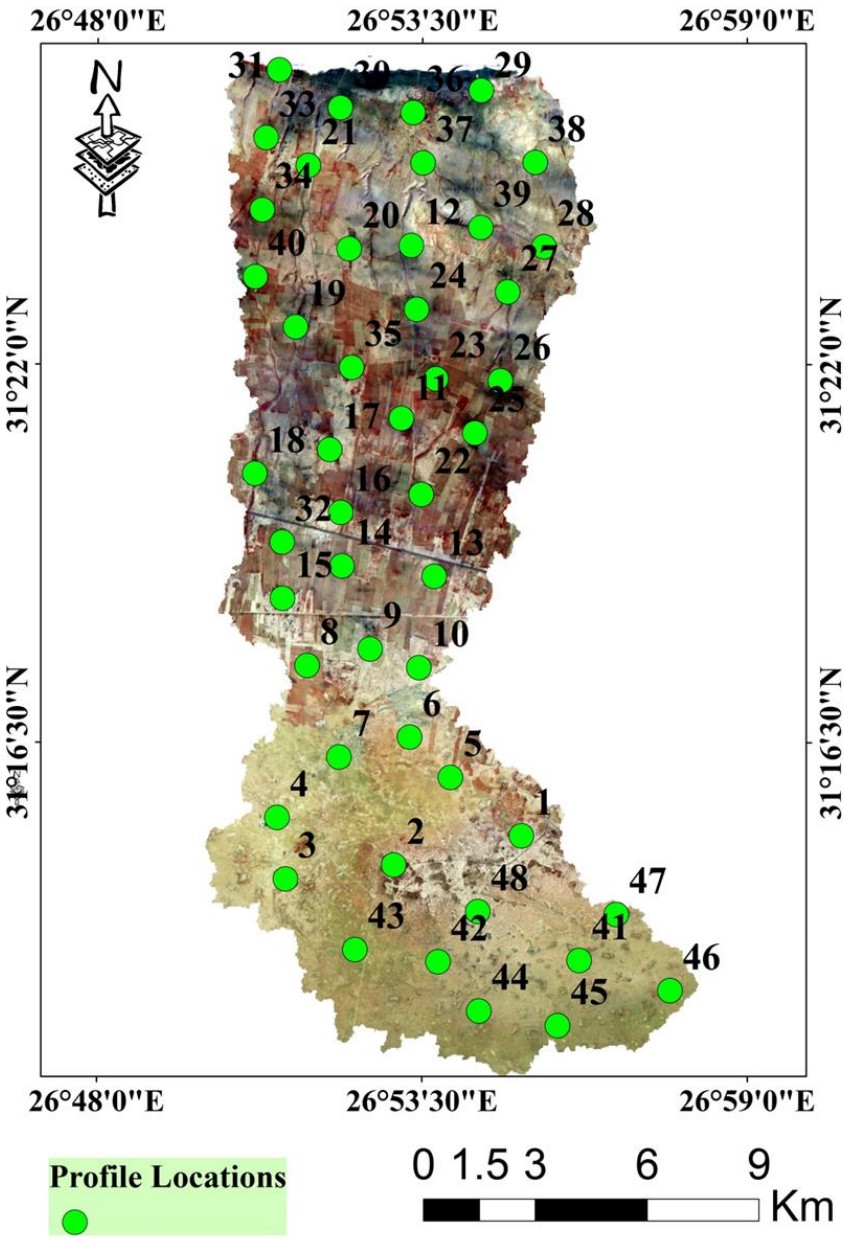

**Figure 3.** Profile distribution within study area.

### 2.6. Assessment of Soil Quality

The different agricultural practices and various types of land use influence the physical, chemical, and biological properties, affecting soil quality [51]. Soil quality index is a flexible model so, it can be used for assessing of soil quality and describe the soil degradation in a specific area [52,53]. In this work, a soil quality index is developed to combine chemical, physical, fertility, and geomorphologic properties of study area for better and accurate estimation of soil quality. Four indices were used to assess soil quality, i.e., geomorphologic (GI), fertility (FI), physical (PI), and chemical (CI), as the following:

The GI was described according to Equation (7):

$$GI = (G_S \times G_A \times G_{PC} \times G_{PrC})^{1/4} \tag{7}$$

where: GI = geomorphologic index; $G_S$ = slope; $G_A$ = aspect; $G_{PC}$ = plan curvature; and $G_{PrC}$ = profile curvature.

The four mentioned variables integrated in the GI indicate which lands can be exposed to erosion and, implicitly, to degradation [45]. In the standard MEDALUS model the slope is integrated in soil quality index (SQI) a parameter integrated as indicator to intensity changes in hydric erosion [54]. The slope aspect conditions expose lands to both water and wind erosion [55–57]. Plan curvature, which indicates the horizontal degree of slope curvature focus on convergent runoff sectors [58,59], soil losses through erosion created different types of SQ-related problems [60]. Thus, it is included in this work for first time to reflect SQ status in the investigate area accurately.

The fertility index was described according to Equation (8):

$$FI = (F_N \times F_P \times F_K \times F_{OM} \times F_{NDVI})^{1/5} \tag{8}$$

where: FI = fertility index; $F_N$, $F_P$, $F_K$ = available nitrogen, phosphorus, potassium, respectively; $F_{OM}$ = organic matter (%); and $F_{NDVI}$ = Normalized Difference Vegetation Index. NDVI added to fertility index as it is sensitive to vegetation conditions dynamic change, including several factors, for instance soil quality [61,62] in addition, there are positive correlation between NDVI and soil quality [2].

The physical index was described according to Equation (9):

$$PI = (P_D \times P_T \times P_{Bd} \times P_{HC} \times P_{WHC} \times P_S)^{1/6} \tag{9}$$

where: PI = physical index; $P_D$ = profile depth; $P_T$ = soil texture; $P_{BD}$ = bulk density (g/cm$^3$); $P_{HC}$ = hydraulic conductivity (cm/h); $W_{HC}$ = water holding capacity (%); and Ps = % surface stoniness.

The chemical index was described according to Equation (10):

$$CI = (C_{EC} \times C_{pH} \times C_{CaCO3} \times C_{CaSO4} \times C_{ESP} \times C_{CEC})^{1/6} \tag{10}$$

where: CI = chemical index; $C_{EC}$ = soil salinity; $C_{pH}$ = soil reaction; $C_{CaCO3}$ = proportion of soil calcium carbonate; $C_{CaSO4}$ = prcentage of gypsum $C_{ESP}$ = soil exchangeable sodium percentage; and $C_{CEC}$ = cation exchange capacity.

The final DSQM index was described according to Equation (11):

$$DSQM = (CI \times PI \times FI \times GI)^{1/4} \tag{11}$$

where: DSQM = Developed soil quality model; CI = chemical index; PI = physical index; FI = fertility index; and GI = geomorphologic index. The parameters or factors were rated (Tables S1–S4) based on experts' suggestions and a review of literature [15,54,63–68].

The NDVI values ranges from −1 to +1 [69], negative values express bare surface, water and clods while positive values represent vegetated surfaces [70,71]. Sentential 2 data have the ability to distinguish vegetation cover [72].

## 2.7. Modeling of Soil Quality Parameters

To produce the spatial model of SQ, the model builder function in ArcGIS 10.7 was used. This tool displays selected spatial analysis of parameters in a diagram chain [2] (Figure 4). Output of each process is used as the input to next process. The following stages were implemented in this work to obtain the final SQ map of the study area: (a) soil properties were interpolated from point based to raster layer; (b) the output from (a) reclassified into five classes (i.e., very high, high, moderate, low, and very low); (c) assigning score for each SQ parameter according to (Tables S1–S4) feeding Equations (7)–(11) using the raster calculator tool; and I the output from (d) used as input in weighted overlay function to produce and display the DSQM final map.

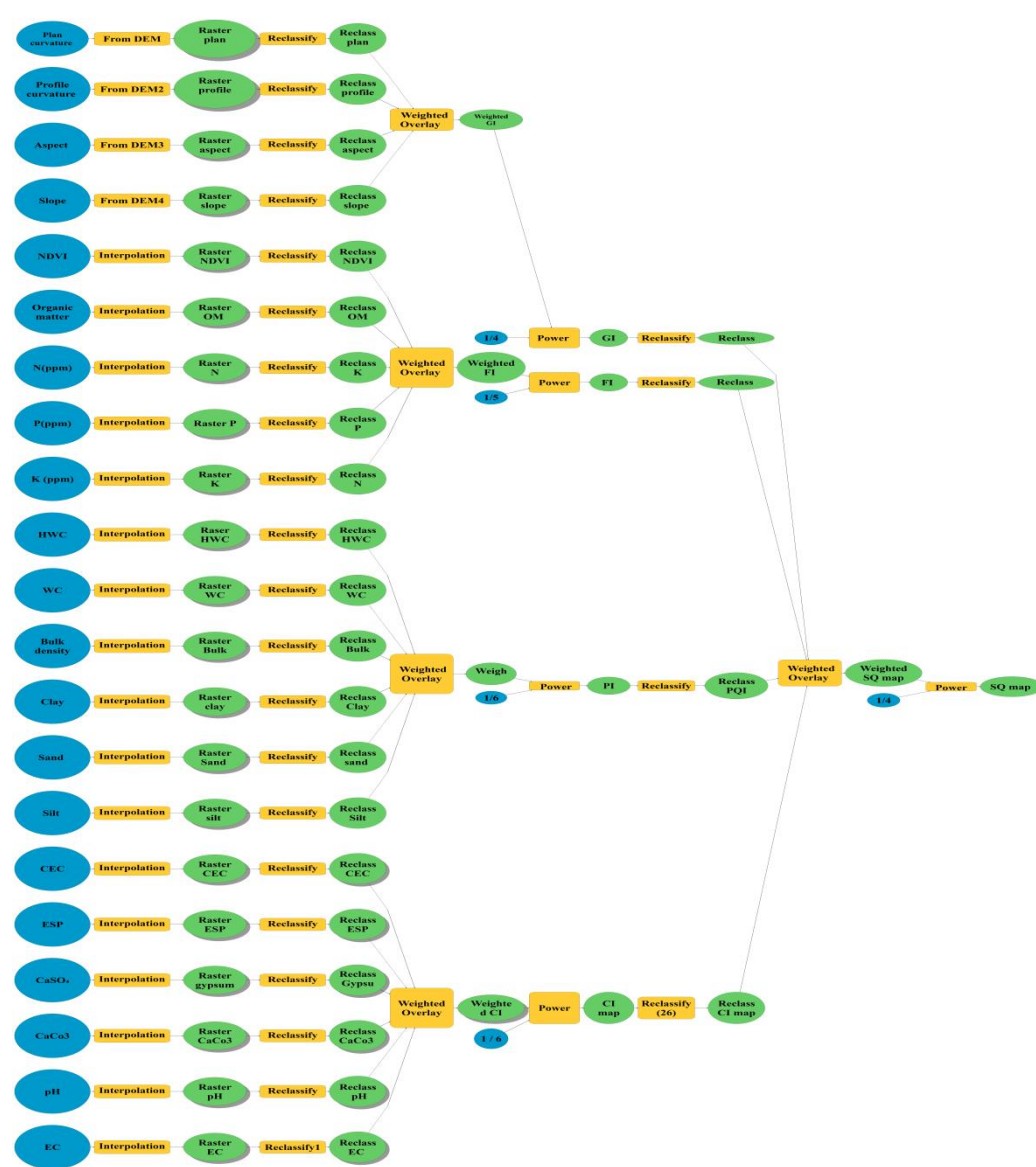

**Figure 4.** Structure of developed model based on model builder in ArcGIS.

## 2.8. Validation of Developed Model

### 2.8.1. Validation Using Kappa Analysis

Weighted Additive method was used according to the following Equation (12) for validation:

$$\text{WAI} = \sum_{i=1}^{n} \text{Wi} \times \text{Si} \tag{12}$$

where: WAI = Weighted Additive index; Si = the indicator score; n = number of indicators; and Wi = the weight of indicators.

All parameters were weighted according to communality of each indicator which calculated by mathematical statistics means of factor analysis using (IBM, SPSS Statics 22). The calculation of weight value for each parameter depended on divided each value by summation of overall values, on other words, as a ratio [73]. Kappa coefficient was used to assess the level of agreement between model and weighted additive model. It is a quantitative measure of consistency for two rates that are rating the same thing according to the following Equation (13):

$$k = I \tag{13}$$

where: K = the kappa coefficient; P(A) = the percentage of times that the coders agreIand P(E) = the percentage of times that we would expect them to agree by chance. Levels of agreement are shown in (Table 1).

**Table 1.** Interpretation of kappa coefficient results according to [74,75].

| Values | Level of Agreement |
|:---:|:---:|
| $\leq 0$ | no agreement |
| 0.01–0.20 | none to slight |
| 0.21–0.40 | Fair |
| 0.41–0.6 | Moderate |
| 0.61–0.8 | substantial |
| 0.81–1 | almost perfect |

2.8.2. Correlation Based on Land Capability

The ALESarid software was used for evaluation of study area land capability. The output values of this model were correlated with SQ values which extracted from developed using (IBM, SPSS Statics 22) according to the following Equation (14) at a significance level (*p*-value) < 0.001:

$$R^2 = 1 - \frac{\sum_{i=1}^{n}(y_i - \hat{y}_i)^2}{\sum_{i=1}^{n}(y_i - \overline{y_i})^2} \tag{14}$$

where: n = the number of samples; $y_i$ = measured value; $\hat{y}_i$ = the predicted value; and y = the mean of measured value.

*2.9. Evaluation of Geostatistical Analysis*

The above-mentioned geostatistical models were used to map soil properties. Four indices (Equations (15)–(18)) were used to evaluate the models as the following:

$$\text{Mean standardized error (MSE)} = \frac{1}{N}\sum_{i=1}^{N}[Z_1(X_1) - Z_2(X_2)] \tag{15}$$

$$\text{Average standard error (ASE)} = \sqrt{\frac{1}{N} + \sum_{I=1}^{N}\left[Z_1(x_i) - [\sum_{i=1}^{N}Z_2(x_i)]/N\right]^2} \tag{16}$$

$$\text{Root mean square error (RMSE)} = \sqrt{\frac{1}{N}\sum_{I=1}^{N}[Z_1(x_i) - Z_2(x_i)]^2} \tag{17}$$

$$\text{Root mean square standardized error (RMSSE)} = \sqrt{\frac{1}{N}\sum_{I=1}^{N}[Z_1(x_i) - Z_2(x_i)]^2} \tag{18}$$

where: $Z_1(x_i)$ = Measured values; $Z_2(x_i)$ = Expected values.

**3. Results and Discussion**

*3.1. Geomorphology of Study Area*

As shown in Table 2 and Figure 5 plains landscape includes five landforms' units, i.e., sand sheet, sand plain, coastal plain, sand beach, and sabkha (wet and dry) with an area

1978.13, 462.27, 193.85, 113.70, and 705.73 hectares, respectively, it formed by erosion of the plateau. Soils of this unit are very important for agriculture due their flatness. Wadi unit consider one of the most diagnosed geomorphological units in the study area and extends in large areas of the north-western coast of Egypt [32] and occupying the north part of the study area. It receives high amounts of runoff in comparison to surrounding upland due to it is location in gentle slopes. This landscape covers about 733.05 hectare (3.34%) of total area including two landforms' units (wadi and wadi outlet). Terraces units are formed by alluvial sediments cyclic erosion and deposition stages (cut and fill) in a setting that generates a staircase [76]. This unit classified into very high alluvial terraces (1227.42 ha), high alluvial terraces (1241.03 ha), slightly moderate alluvial terrace (1615.76 ha), moderate alluvial terrace (885.84 ha), and low alluvial terraces (2179.60 ha). Basins are defined as lowland where the accumulation of rainfall and drained water done on their outlet. Basins include both of the accumulative surface runoff, and nearby streams which, downslope towards the shared outlet represent 399.27 ha of total area. Pavement plain unit occupies 3588.68 ha (16.8%) of the total area and formed by the erosion processes over a long time. This unit is divided into Pediment plain (high, moderate and low with areas 444.14, 659.26 and 1054.62 hectares, respectively, and peneplain (1430.66 ha). The included landforms in the reference's terms landscape are plateau (1444.62 ha), escarpment (774.97 ha), table land (3444.89), waterlogging, and rock outcrop (380.94 ha).

**Table 2.** Areas of landforms units in the study area.

| Land Scape | Geomorphology | Landforms | Area—km² | Area—Hectare (ha) | % Area |
|---|---|---|---|---|---|
| | Sand sheet | High sand sheet | 19.78 | 1978.13 | 9.26 |
| | Sand plain | Sand plain | 4.62 | 462.27 | 2.16 |
| Plain | Coastal plain | Coastal plain | 1.94 | 193.85 | 0.91 |
| | Sand Beach | Sand Beach | 1.14 | 113.70 | 0.53 |
| | sabkha | Dry sabkha | 6.32 | 631.81 | 2.96 |
| | | Wet sabkha | 0.74 | 73.92 | 0.35 |
| Basin | basin | Basin | 3.99 | 399.27 | 1.87 |
| Wadi | Wadi | Wadi | 7.26 | 726.14 | 3.40 |
| | Wadi | Wadi outlet | 0.07 | 6.91 | 0.03 |
| | | Very High Alluvial terraces | 12.27 | 1227.42 | 5.74 |
| | | high Alluvial terraces | 12.41 | 1241.03 | 5.81 |
| Terraces | Alluvial terraces | Slightly Moderate Alluvial terrace | 16.16 | 1615.76 | 7.56 |
| | | Moderate Alluvial terrace | 8.86 | 885.84 | 4.15 |
| | | Low Alluvial terraces | 21.80 | 2179.60 | 10.20 |
| | | High Pediment plain | 4.44 | 444.14 | 2.08 |
| Pavement plain | Pediment plain | Moderate Pediment plain | 6.59 | 659.26 | 3.09 |
| | | Low Pediment plain | 10.55 | 1054.62 | 4.94 |
| | peneplain | Peneplain | 14.31 | 1430.66 | 6.69 |
| | Plateau | Plateau | 14.45 | 1444.62 | 6.76 |
| | Escarpment | Escarpment | 7.75 | 774.97 | 3.63 |
| | | High table land | 13.07 | 1306.70 | 6.11 |
| Reference terms | Table land | Moderate table land | 13.75 | 1374.56 | 6.43 |
| | | Low table land | 7.64 | 763.63 | 3.57 |
| | Waterlogging | Waterlogging | 0.54 | 54.24 | 0.25 |
| | Rock outcrop | Rock outcrop | 3.27 | 326.70 | 1.53 |

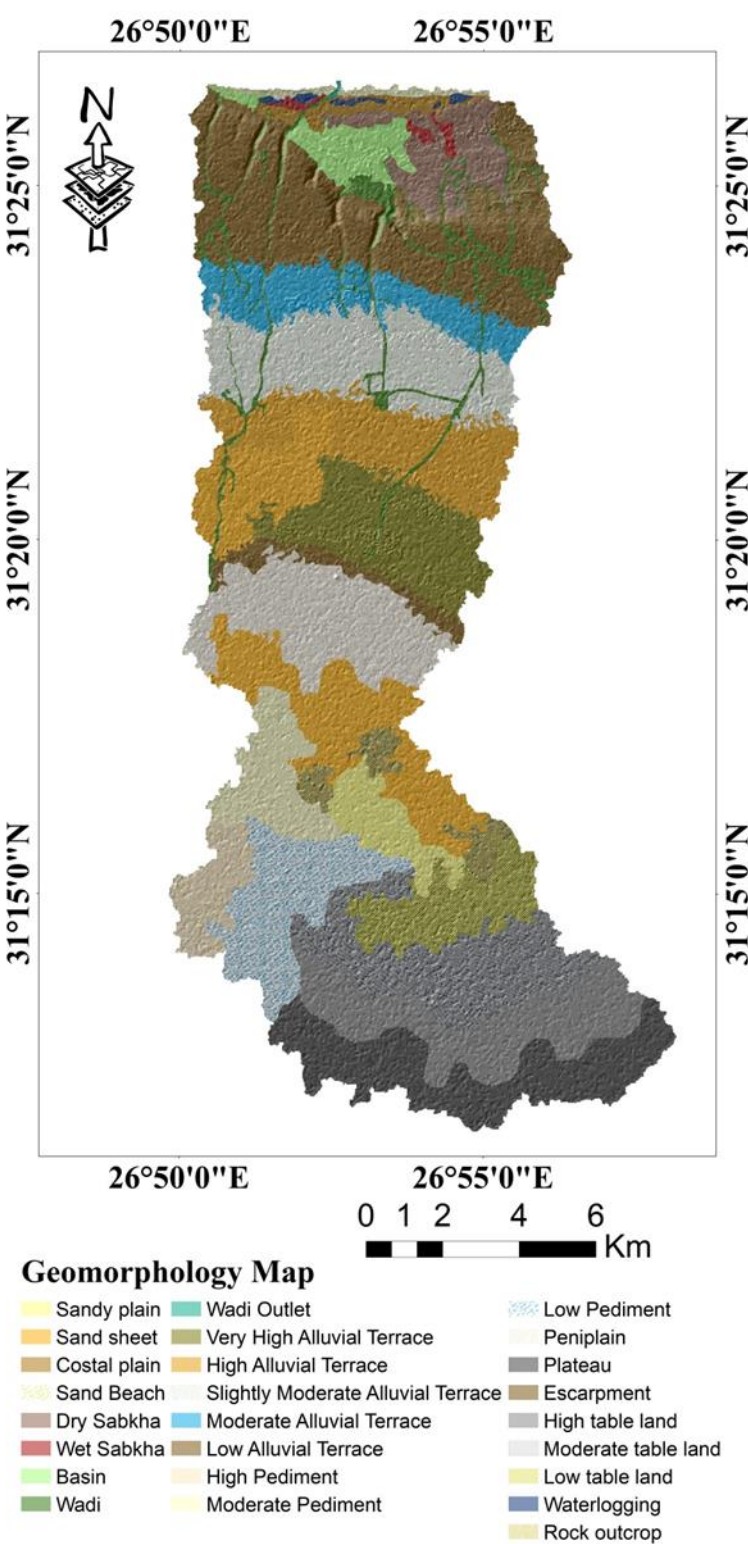

**Figure 5.** Geomorphology map of study area.

### 3.2. Soil Properties of Study Area

Soil properties of the study area are recorded in (Table 3) and interpolated in (Figure S4). The settings of the study area are mildly/strongly alkaline due to the pH values range from 7.90 to 8.54, with an average value of 7.28 [77]. Physical, chemical, and biological properties are affected by soil pH [78,79]. Values of ESP differ from 7.9% to 10.2%, which means that the area is not exposed to sodicity risks [80]. The results of CEC ranged from

1.2 to 7.0 cmole/kg. The low values of it due to low content of clay and organic matter as there are significant positive correlations between CEC, clay, and organic matter [28], while organic matter percentage (OM%) content ranges from 0.04% to 0.71%. There is no specific distribution pattern of OM in the study area (Figure S4d) while, soils of the study area were characterized as having low content of OM, in agreement with [71]. OM play very important role in improving soil physical and chemical properties [81,82]. Texture differs between sandy clay loam, sandy loam, and sandy. Hydraulic conductivity (HC) expresses of water movement and pore structure in soil [83], it ranged from 1.61 to 15.66 (cm/h) in addition water holding capacity is relatively low due to coarse texture. Hydraulic conductivity is an important indicator of water movement and pore structure in soil. Soil depth ranged between 40 and 90 cm while surface stoniness is less than 20%. The study area is diagnosed by none to high salinity soils due to ECe values varying from 0.1 to 14.89 dS/m with an average value of 5.04 dS/m [84]. According to the interpolated map the highest value of EC was found in the north part of study area it might due to sea water (Figure S4k). Most salinized soils are present in arid and semi-arid conditions due to low precipitation and high evaporation [2]. $CaCO_3$ ranges between 2.12% and 34%. Areas near to sea had the highest value of $CaCO_3$; it might be due to shell fragments. The highest $CaCO_3$ value in some areas can cause formation of very hard layers impermeable to water and crop roots in addition, phosphorus fixation fertilizer in calcareous soils [85] areas in the southeast of study area has the highest values of $CaCO_3$ it might due to shell fragments (Figure S4k). $CaSO_4$ ranged from 0.2% to 3.1% with an average 0.6%. The available N ranges from 20.3 to 66.14 mg/kg (45.4 to 148.15 kg N/ha) demonstrating that the nitrogen content in the study area differ from low to moderate [71] while, high content of N located in the middle part of study area due to agriculture practices (Figure S4n). The available P and K content in the study area are classified as low according to [77] as the average values are 9.18 kg P/ha and 186.74 kg K/ha. To estimate and map the unknown soil properties, the OK interpolation method was used (Figure S4). Accuracy of the model was confirmed for each soil property depending on mean standardized error (MSE), average standard error (ASE), root mean square error (RMSE), and root mean square standardized error (RMSSE) (Table 4). The results indicate that spherical model is suitable for EC, pH, bulk density, HC, WHC, and $CaSO_4$, the Gaussian model is suitable for OM, ESP, sand% N, P, and K. Finally, the circular and exponential models are suitable for K and clay%, as RMSSE and MSE are close to one and zero, respectively, thus the mentioned models are the appropriate for predicting the unsampled location [67,86].

**Table 3.** Statistics of some soil properties which used in SQ assessment.

| Properties | Min. | Max. | Mean | Standard Division (STD) |
|---|---|---|---|---|
| pH | 7.90 | 8.54 | 7.28 | 0.35 |
| ESP | 7.93 | 12.61 | 10.24 | 1.03 |
| CEC (cmole/kg) | 1.19 | 6.98 | 3.46 | 1.68 |
| OM% | 0.05 | 0.71 | 0.38 | 0.17 |
| Bulk density (g/cm$^3$) | 1.12 | 1.68 | 1.48 | 0.16 |
| HC (cm/h) | 1.61 | 15.66 | 11.09 | 3.25 |
| WHC (%) | 10 | 20 | 10.79 | 2.43 |
| EC (dS/m) | 0.10 | 14.89 | 5.04 | 3.67 |
| CaCO$_3$ (%) | 2.12 | 34 | 12.59 | 5.10 |
| CaSO$_4$ (%) | 0.22 | 3.10 | 0.62 | 0.49 |
| N (ppm) | 20.3 | 45.47 | 66.15 | 9.63 |
| P (ppm) | 2.01 | 6.84 | 4.18 | 1.40 |
| K (ppm) | 24.13 | 174.59 | 83.37 | 30.64 |

**Table 4.** Semi-variogram models and geostatistical analyses of some soil properties.

| Soil Parameters | Model Type | Mean | RMSE | MSE | RMSSE | ASE |
|---|---|---|---|---|---|---|
| pH | Spherical | 0.03 | 0.40 | 0.07 | 0.91 | 0.38 |
| ESP | Gaussian | 0.0003 | 0.93 | 0.00 | 1.32 | 0.69 |
| CEC (cmole/kg) | Circular | 0.003 | 1.65 | 0.00 | 1.00 | 1.64 |
| OM% | Gaussian | 0.002 | 0.15 | 0.01 | 0.97 | 0.16 |
| Bulk density (g/cm$^3$) | Spherical | 0.002 | 0.1687 | 0.01 | 0.99 | 0.16 |
| HC (cm/h) | Spherical | 0.46 | 3.13 | 0.07 | 0.56 | 5.97 |
| WHC (%) | Spherical | 0.03 | 2.43 | 0.005 | 1.16 | 1.97 |
| EC (dS/m) | Spherical | 0.22 | 2.65 | 0.01 | 0.87 | 3.95 |
| CaCO$_3$ (%) | Gaussian | 0.01 | 9.40 | 0.00 | 0.98 | 9.40 |
| CaSO$_4$ (%) | Spherical | 0.01 | 0.48 | 0.04 | 1.1 | 0.40 |
| Sand (%) | Gaussian | 0.15 | 6.34 | 0.04 | 0.97 | 6.76 |
| Silt (%) | Gaussian | 0.52 | 4.36 | 0.02 | 1.15 | 1.93 |
| Clay (%) | Exponential | 0.15 | 0.99 | 0.03 | 0.99 | 4.32 |
| N (ppm) | Gaussian | 5.31 | 14.08 | 0.1 | 0.49 | 28.02 |
| P(ppm) | Gaussian | 0.009 | 1.24 | 0.02 | 0.94 | 1.36 |
| K (ppm) | Gaussian | 0.77 | 26.72 | 0.03 | 1.02 | 25.24 |

### 3.3. Geomorphological Index (GI)

Geomorphologically, the GI values indicate that about 307.55 ha is located under very high class (G1) while around 9867 ha is high class (G2) and moderate geomorphology class (G3) cover 6919.26 ha of total study area and the poorest geomorphological conditions, i.e., low (G4) covers around 3045.7 ha due to the values of slope, aspect, plan, profile curvature show an accelerated, convergent surface runoff leads to hydric erosion (Figure S3a–d; Tables S5, S6, and Table 5 and Figure 6).

**Table 5.** Areas of geomorphologic index.

| Classes | Symbol | Area (ha) |
|---|---|---|
| Very high | G1 | 307.55 |
| High | G2 | 9867.42 |
| Moderate | G3 | 6919.26 |
| Low | G4 | 3045.671 |
| | Reference terms | 1229.82 |

### 3.4. Fertility Index (FI)

Soil fertility mapping is a key issue for a lot of implementations in research fields ranging from sustainability of soil management to the precision farming concept [67]. According to FI index, the study area fell into very high (F1), high (F2), moderate (F3), and low (F4) classes, respectively. A descending order of fertility index in the study area is F3 (9946.33 ha), F4 (7729.22 ha), F2 (2350.65 ha), and F1(113.70 ha) (Table 6 and Figure 7). As clear from results that most of study area located under F3 class because of deficiency of OM, N, P, and K values in addition low values of NDVI from remote sensing data over the study. NDVI is helping in achieving precision agriculture by predicting and mapping the land degradation extension and allowing farmers and decision makers to make accurate decisions on time [2,87]. The major causes of low productivity are soil fertility losing and nutrients depletion [88], so to achieve sustainable development, it requires decreasing losing and increasing the efficiency of use [89].

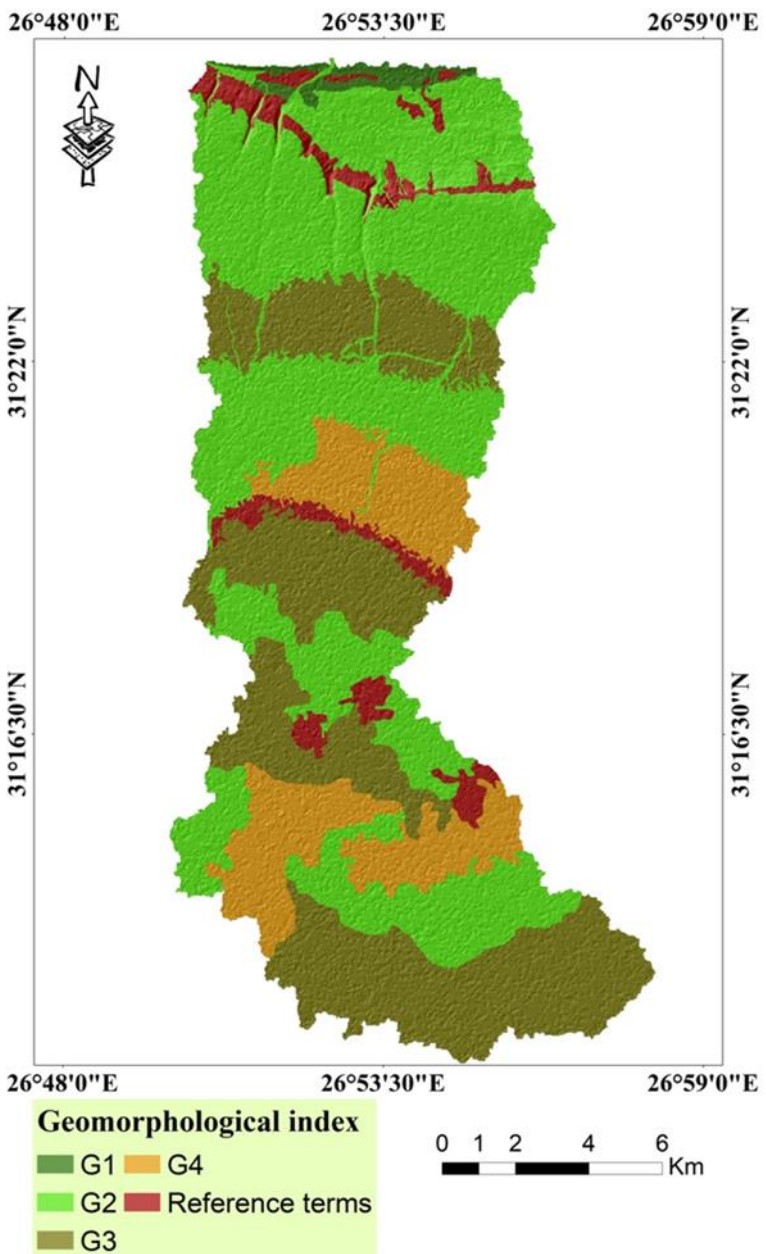

**Figure 6.** Spatial distribution of GI in the study area.

**Table 6.** Areas of fertility index.

| Classes | Symbol | Area (ha) |
|---|---|---|
| Very high | F1 | 113.70 |
| High | F2 | 2350.65 |
| Moderate | F3 | 9946.33 |
| Low | F4 | 7729.22 |
| | Reference terms | 1229.82 |

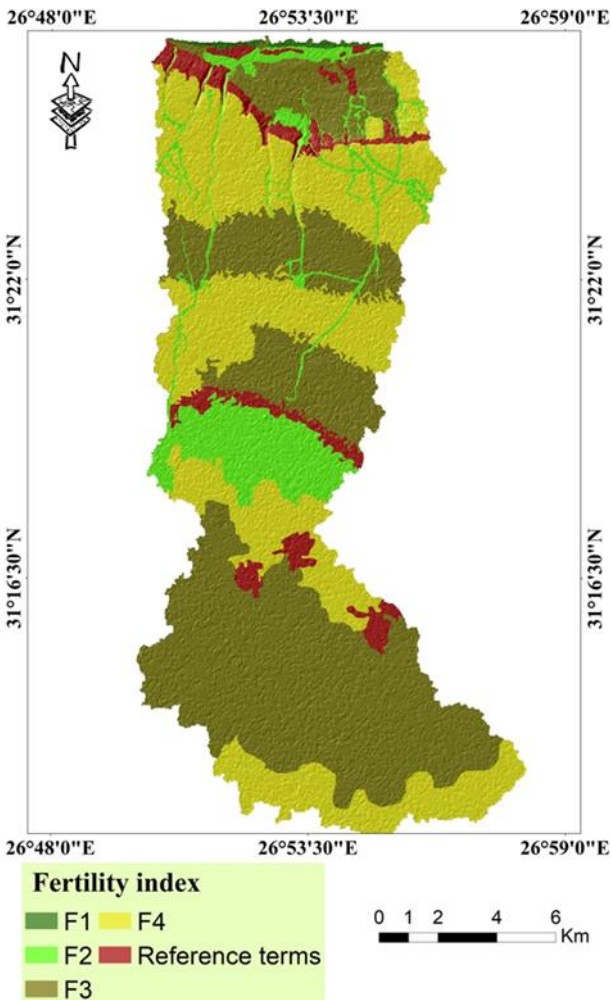

**Figure 7.** Spatial distribution of FI in the study area.

*3.5. Physical Index*

The data given in Table 7 and Figure 8 and indicate that PI in the study area is varied from very high physical index (P1) to very low (P5). Rating of soil physical index indicates that around 7 (ha) of study area is very high-quality soil due to deep soil profiles, low content of gravels, while 6457.6, 6634.14, 6409.43, and 331.80 (ha) of total agriculture areas are classified as high (P2), moderate (P3), low (P4), and very low (P5) classes, respectively. The soil limiting factors are coarse texture, high values of bulk density, and shallow depth.

**Table 7.** Areas of physical index.

| Classes | Symbol | Area (ha) |
|---|---|---|
| Very high | P1 | 6.90 |
| High | P2 | 6457.61 |
| Moderate | P3 | 6634.14 |
| Low | P4 | 6409.4323 |
| Very low | P5 | 631.80 |
| | Reference terms | 1229.82 |

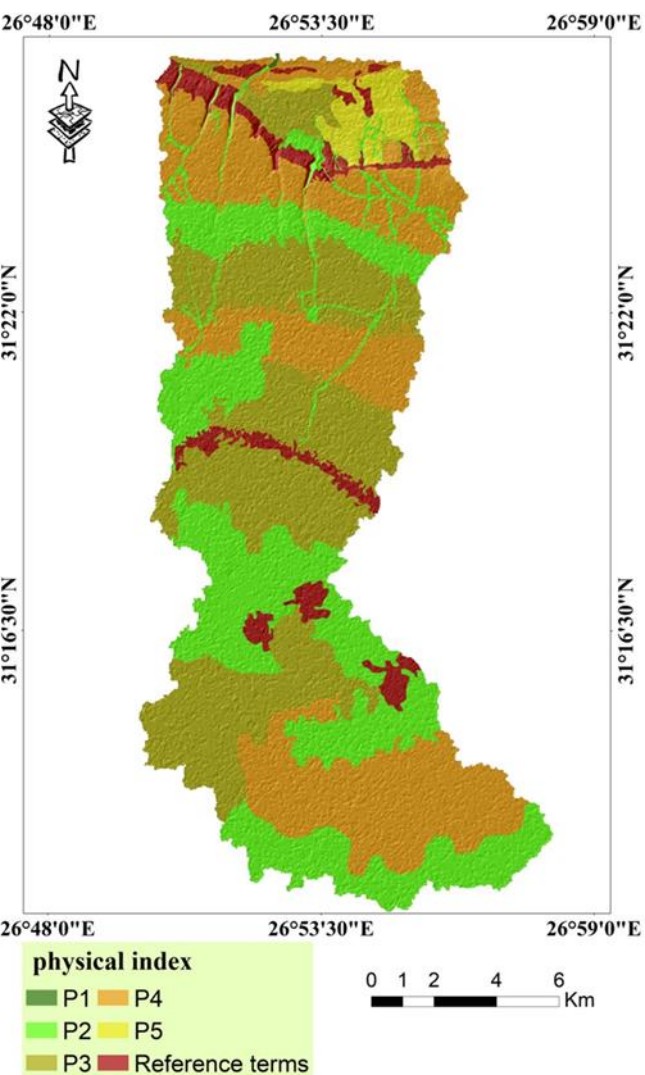

**Figure 8.** Spatial distribution of physical index in the study area.

*3.6. Chemical Index (CI)*

Soil degradation (chemical, physical, and biological), means reducing of soil quality [89]. Spatial distribution map of chemical index (Figure 9) shows that a wide range of chemical quality ranging from very low quality (C5) to very high (C1). The areas chemical index are as the follows: 7291.9 ha is very high quality, 3794.7 ha is high quality, 3407 ha is moderate quality, 3773.91 ha is low quality, and 1872.83 ha is very low. C1 class characterized by low values of ECe, ESP, and pH on the other hand, low values of CEC which may causing chemical degradation [90] and high content of $CaCO_3$ are the main limiting factors of C5 class soils (Table 8).

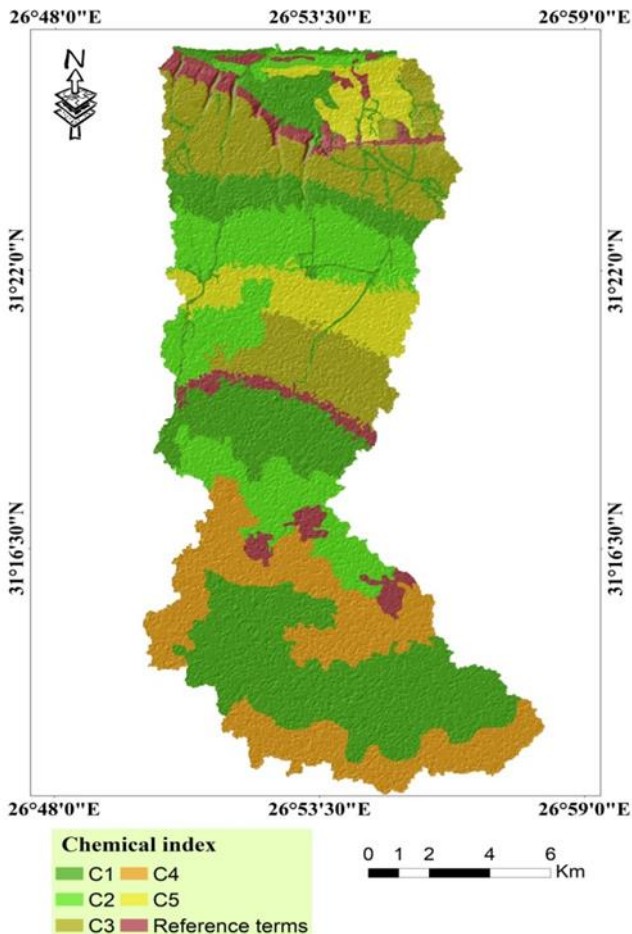

**Figure 9.** Spatial distribution of CI in the study area.

**Table 8.** Areas of chemical index.

| Classes | Symbol | Area (ha) |
|---------|--------|-----------|
| Very high | C1 | 7291.49 |
| High | C2 | 3794.64 |
| Moderate | C3 | 3407.02 |
| Low | C4 | 3773.91 |
| Very high | C5 | 1872.83 |
| | Reference terms | 1229.82 |

### 3.7. Assessment of Soil Quality

Soil physical, chemical, and biological quality parameters are the key indicators of SQ. An optimal combination of these parameters increases agronomic productivity and reach to management systems sustainability [91], furthermore geomorphologic properties have a direct effect on land state through increasing of hydric erosion process [51]. According to (Figure 10) the DSQM the study area classified into four classes. The first class is characterized by high quality represent around 194 ha (0.9%) of the total study area. The second class is characterized by moderate quality occupied 4748.61 ha (22.22%) of the total study area. The soil third quality class (low) covers 4675 ha (21.87%) of the total study area and, finally, the very low-quality class is the most representative class as it occupies 10522.45 ha (49.23%) of total area. Around 5.7% of study area is reference, i.e., table land, waterlogging and rock outcrop, these areas are not cultivated. It could be concluded from the interpolated map that; the highest class of soil quality is located in the wadi unit (Figure 10a). Low values of OM%, CEC, N, P, K led to negative effect on soil

quality in addition physical properties, i.e., shallow depth, coarse texture affect particles and pores organization and therefor, impacts on root growth, speed of plant emergence, and agricultural practices water infiltration [83].

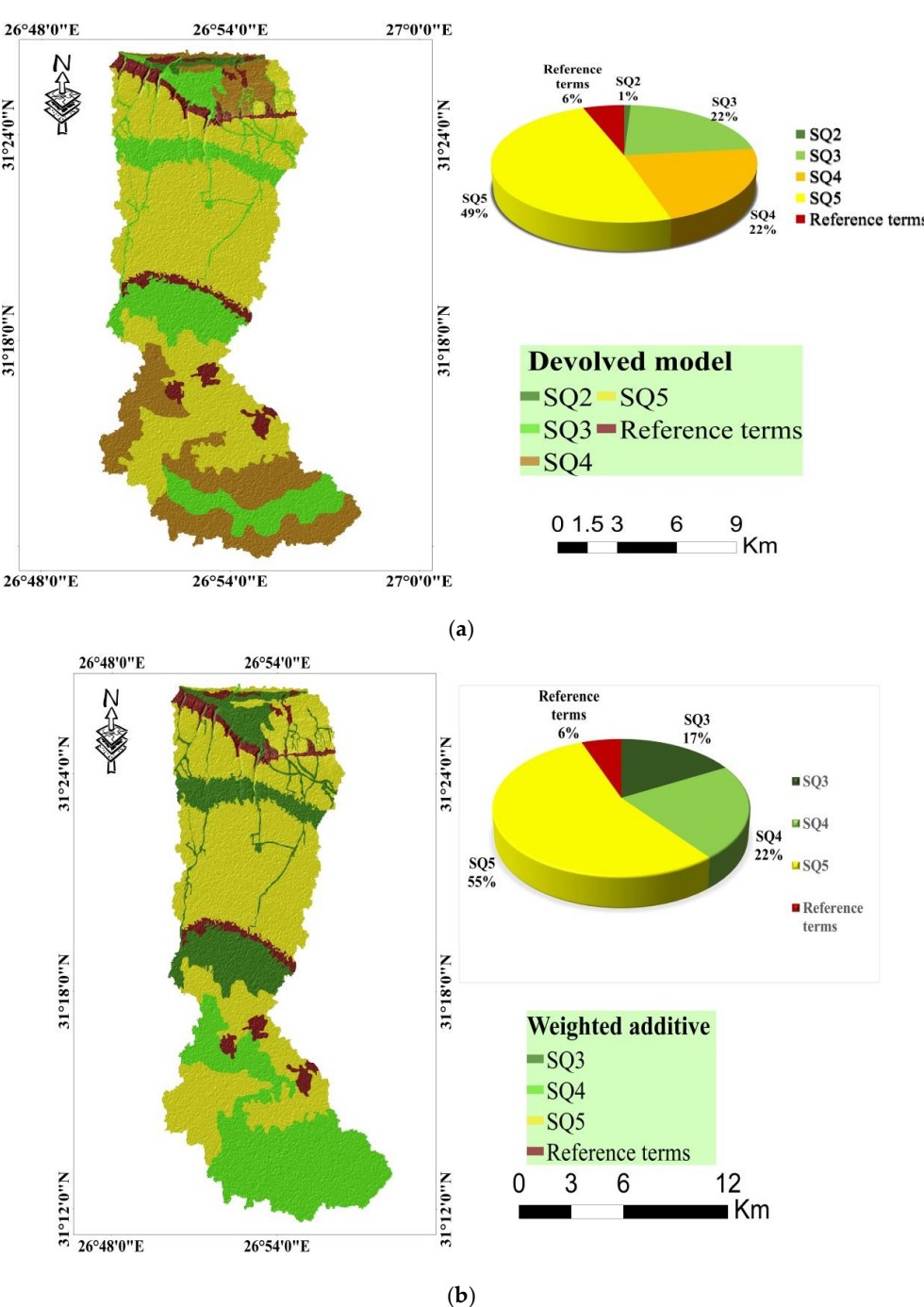

**Figure 10.** Spatial distribution of SQ in the study area: (**a**) developed soil quality model (DSQM); and (**b**) weighted additive.

Generally, soil quality is affected by agricultural practices and climatic conditions, which, in turn, affect the physical, chemical, and fertility properties of the soil [25]. Calculation of SQ in the study area by weighted additive index were according to the following Equation (19):

$$SQ = (0.047 \times SS) + (0.038 \times SA) + (0.047 \times PC) + (0.03 \times SPRC) + (0.036 \times SN) \times$$
$$(0.042 \times SP)(0.044 \times SK) + (0.024 \times SOM) + (0.037 \times SNDVI) + (0.039 \times SD) +$$
$$(0.046 \times ST) + (0.047 \times SBD) + (0.05 \times SHC) + (0.043 \times SWHC) + (0.04 \times SSS) +$$
$$(0.004 \times SEC) + (0.047 \times SpH) + (0.045 \times SCaCO3) + (0.046 \times SCaSO4) +$$
$$(0.04 \times SESP) + (0.037 \times SCEC) \tag{19}$$

The results shows that 3635.76 ha (17%) of soils are moderate quality, 4792 ha (22.4%) of soils are fell into low class, and while, most of study area are characterized by very low class around 11,712.11 ha (54.8%) (Figure 10b). These results indicate high agreement between weighted additive index and developed model as most of study area around 49.23% is classified as very low-quality class according to developed model.

*3.8. Assessment of Land Capability*

The results of land capability are showed in (Figure 11) it could be concluded from these results that the study area fell into three classes, i.e., fair (C3), poor (C4), and non-agriculture class (C5) with an area of 1125.41 ha (5.26%), 12,713.27 ha (59.50%), and 6301.22 ha (29.50%), respectively. Moreover, around 6% of study area is classified as references terms (uncultivated area). The results showed that the most of study area is classified as poor for agriculture (59.50%) which means that these soils have a lot of hazards, such as shallow depth, coarse texture low values of fertility indicators. The lowest class located in the north part of study area and some areas near to the middle of it (Figure 10). The soils of C5 cannot be cultivated consistently; due to the management process of agriculture is difficult [61].

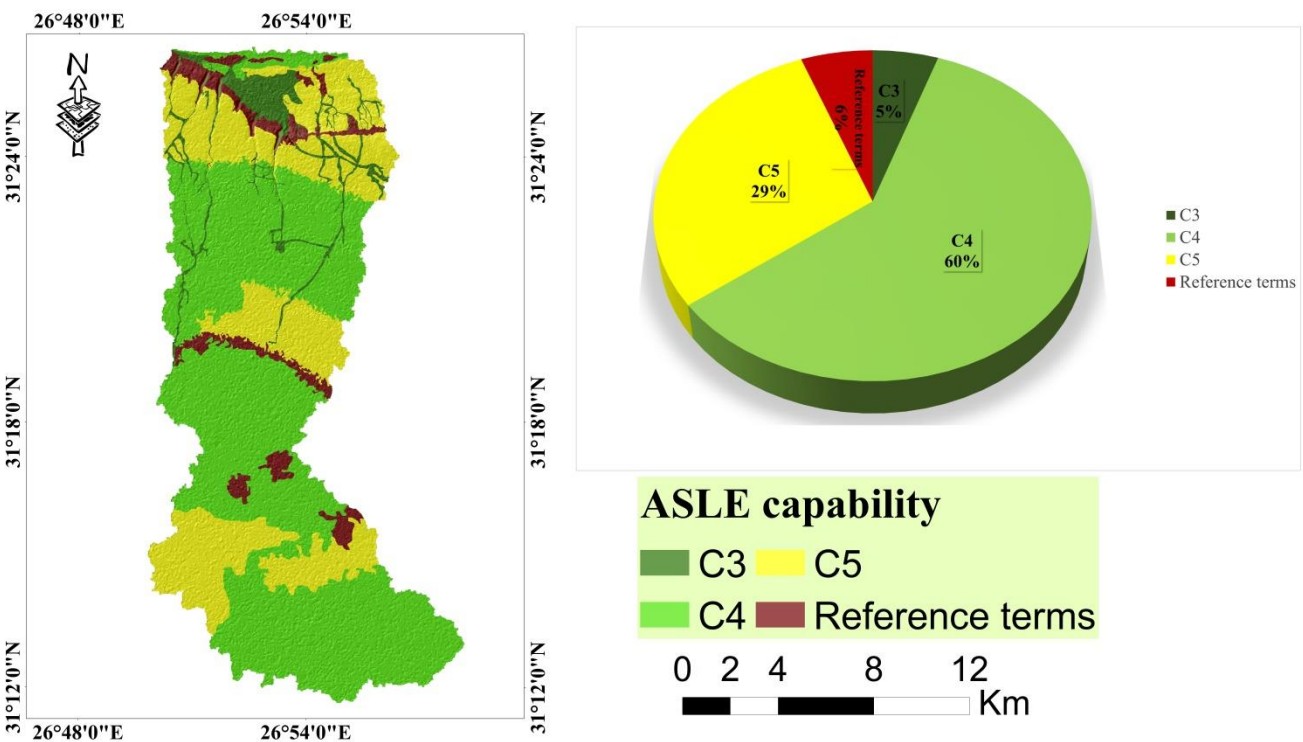

**Figure 11.** Spatial distribution of LC in study area.

*3.9. Developed Soil Quality Model (DSQM) Validation*

To calculate the agreement between developed model and weighted additive index, kappa coefficient was calculated. The kappa value was 0.67 indicating substantial agreement (Table 1). It is implemented to show agreement between the two models involves all

parameters regardless their relative importance or weights [18]. Although, the geometric mean algorism provides enhanced assessment of central conditions due to the arithmetic mean may be skewed away from the median because the presence of outliers and anomalous results [22]. DSQM is s significantly high correlated with CI ($R^2$ = 0.71, $p$ < 0.001), these results indicate that the DSQM is accurate model for assessment of soil quality in the agreement of [2]. The CI is chosen because it is using for assessing the potential of land for a specific type of use like DSQM, but the DSQM is a newly developed approach, while CI has been in use since 1961 [92].

## 4. Conclusions

Generally, soil quality assessment is very important for sustainable agricultural practices management and for precision farming especially. In this study soil physical, chemical, biological, and geomorphological properties were used for SQ evaluation. For this purpose, GMA jointly used with GIS to quantitative assessment of SQ and map it. The results indicated that the soil quality (SQ) of study area classified into four classes, i.e., high quality (SQ2), moderate quality (SQ3), low quality (SQ4), and very low quality (SQ5) occupied 0.90%, 21.87%, 22.22%, and 49.23% of the total study area in addition 5.74% of study area is uncultivated. The results were validated by calculation kappa coefficient and showed substantial agreement with weighted additive index, moreover significantly high correlated with CI. As a whole, the developed soil quality model (DSQM) is qualified to assess soil quality actuary in the study area and re applied in the same environments. Improving SQ in the study area requires some agriculture practices for instance; reduce risks of soil erosion which might occurs due to, geomorphologic properties of study area, increase the rainwater and fertilizers efficiency used as, the cultivation in the study area depends on winter rains and minimize losing of organic matter and nutrients. In conclusion, it is very important to assess soil quality periodically to identify limiting factors of SQ and try to maintain high crop yield, for the purpose of reduce a gap between production and consumption and it is suggested that increasing field work and approaches of soil quality calculation in the future studies.

**Supplementary Materials:** The following are available online at https://www.mdpi.com/article/10.3390/su132313438/s1. Figure S1: The main geological units of study area. Figure S2: NDVI of investigated area. Figure S3: The indices of topographic in the study area: (a) slope; (b) aspect; (c) plan curvature; (d) profile curvature; (e) Slop length factor (LS-factor); (f) slope length and steepness; (g) valley depth; and (h) analytical hill shading. Figure S4: Interpolation maps of some physical, chemical, and biological properties: (a) soil reaction (pH); (b) exchangeable sodium percentage (ESP); (c) cation exchange capacity (CEC, cmole/kg); (d) organic matter (OM, %); (e) sand (%); (f) silt (%); (g) clay (%); (h) bulk density (g/cm$^3$); (i) hydraulic conductivity (cm/h); (j) water holding capacity (%); (k) soil salinity (EC, ds/m); (l) calcium carbonate ($CaCO_3$, %); (m) gypsum ($CaSO_4$, %); (n) nitrogen (N, ppm); (o) phosphorus (P, ppm); and (p) potassium (K, ppm). Table S1: Scores of GI parameters. Table S2: Scores of FI parameters. Table S3: Scores of PI parameters. Table S4: Scores of CI parameters. Table S5: Values and classes of CI, FI, PI, and GI indices. Table S6: Final SQ range of study area.

**Author Contributions:** Conceptualization, M.A.A., A.A.E.B., E.K.M., A.M.S., F.S.M. and M.S.S.; methodology, M.A.A., A.A.E.B., E.K.M., A.M.S., F.S.M. and M.S.S.; software, M.A.A., A.A.E.B., E.K.M., A.M.S., F.S.M. and M.S.S.; validation, M.A.A., A.A.E.B., E.K.M., A.M.S., F.S.M. and M.S.S.; formal analysis, M.A.A., A.A.E.B., E.K.M., A.M.S., F.S.M. and M.S.S.; investigation, M.A.A., A.A.E.B., E.K.M., A.M.S., F.S.M. and M.S.S.; resources, M.A.; data curation, M.A.A., A.A.E.B., E.K.M., A.M.S., F.S.M. and M.S.S.; writing—original draft preparation, M.A.A., A.A.E.B., E.K.M., A.M.S., F.S.M. and M.S.S.; writing—review and editing, M.A., K.H.S. and E.M.E.; supervision, E.M.E.; project administration, E.M.E.; funding acquisition, M.A. All authors have read and agreed to the published version of the manuscript.

**Funding:** This research was funded by the Deanship of Scientific Research at King Khalid University (grant number RGP.1/301/42).

**Institutional Review Board Statement:** Not applicable.

**Informed Consent Statement:** Not applicable.

**Data Availability Statement:** The data presented in this study are available in the main manuscript and the supplementary materials.

**Conflicts of Interest:** The authors declare no conflict of interest.

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
