# Peer review of "A GIS-Based Approach for the Quantitative Assessment of Soil Quality and Sustainable Agriculture"

_sustainability, doi:10.3390/su132313438_

Round 1
Reviewer 1 Report
This current paper deals with general research tasks. The knowledge is about the “A Developed Approach to the Quantitative Soil Quality Assessment for Potential Agricultural Sustainability based on GIS Modelling”. The paper contains some useful information for local place and for environmental or soil science. As it well known that knowledge of the soil quality index is critical for making decisions with respect to terrestrial ecosystem serves for decision makers to sustain particularly soil and natural land resources. In this current study, statement of manuscript is clarity. The main objectives of this research were given by authors in introduction section and these objectives were clarity explained in result and discussion section. On the other hand, on the world, these kinds of investigations were performed by many researchers using these types of indicators or using such kind of SQI model. The work presented in the manuscript prepares Quantitative Soil Quality assessment for Potential Agricultural Sustainability by combining (simple multiplication, (Eq. (14))) four thematic maps (i.e., GI, FI, PI and CI). It is a simple GIS analysis, i.e., raster calculation. The manuscript lags in terms of novelty. Even for publishing as a case study, the manuscript requires additional analysis. For example, DSQ index can be compared with other indices. Therefore, there is no any new approach or revision or additional innovation in order to develop soil quality model. Moreover, the most important thing is there is no any verification with field data using any crop yield for results of SQI. Just using NDVI approach to validate this model is not enough. In addition, authors already mention that the study area has been used under intensive agriculture so, authors should have also given field experiment data or questionary-survey study or statistical real data by comparing between crop yield and SQI to make check or validate them. As for your conclusion, the results and discussion are insufficient, please describe more about the research results and it's better to make a series of important policy suggestions for policy makers. In addition, you should discuss any weaknesses or points that you have not considered in your paper.
Author Response
- November 2021
Anastasija Milenkovic
Assistant Editor
Sustainability
Dear Anastasija Milenkovic,
Please find attached the revised manuscript titled ‘A Developed Approach to the Quantitative Soil Quality Assessment for Potential Agricultural Sustainability Based on GIS Modelling’. Ms. Ref. No.: sustainability-1454395, authored by Abdellatif, M.A.; El Baroudy, A.A.; Arshad, M.; Mahmoud, E.K.; Saleh, A.M.; Moghanm, F.S.; Shaltout, K.H.; Eid, E.M.; and Shokr, M.S.
On behalf of my co-authors, I thank you very much for giving us the opportunity to revise our manuscript. We appreciate the positive and constructive comments and suggestions provided by the reviewers on our manuscript. We have carefully studied the reviewers’ comments and have made revisions that are tracked in the revised version of the manuscript. We have tried our best to revise our manuscript according to the reviewers’ comments. Please find attached the revised version of our manuscript, which we would like to submit for your kind consideration. Once again, we would like to express our great appreciation to you and the reviewers for the comments on our manuscript.
Please find below our detailed responses to each of the points raised:
---------------------------------------------------------------------------------------------------------------------
Comments of Reviewer # 1:
This current paper deals with general research tasks. The knowledge is about the “A Developed Approach to the Quantitative Soil Quality Assessment for Potential Agricultural Sustainability based on GIS Modelling”. The paper contains some useful information for local place and for environmental or soil science. As it well known that knowledge of the soil quality index is critical for making decisions with respect to terrestrial ecosystem serves for decision makers to sustain particularly soil and natural land resources. In this current study, statement of manuscript is clarity. The main objectives of this research were given by authors in introduction section and these objectives were clarity explained in result and discussion section.
On the other hand, on the world, these kinds of investigations were performed by many researchers using these types of indicators or using such kind of SQI model. The work presented in the manuscript prepares Quantitative Soil Quality assessment for Potential Agricultural Sustainability by combining (simple multiplication, (Eq. (14))) four thematic maps (i.e., GI, FI, PI and CI). It is a simple GIS analysis, i.e., raster calculation. The manuscript lags in terms of novelty. Even for publishing as a case study, the manuscript requires additional analysis. For example, DSQ index can be compared with other indices. Therefore, there is no any new approach or revision or additional innovation in order to develop soil quality model.
Response: Thanks so much Sir for your time and for positive and constructive comments and suggestions. The present study aims to use simple approach to calculate the soil quality and accurate at the same time considering a lot of parameters which effect on soil quality. The novelty of this work is using GI as new index and NDVI as a new parameter integrated with other indices to reflect the soil quality as accurately as possible. Results of DSQ compared with weighted additive model and shows substantial agreement with it. The weighted additive model established by chen et al [67] and used for validation by [15].
---------------------------------------------------------------------------------------------------------------------
Moreover, the most important thing is there is no any verification with field data using any crop yield for results of SQI. Just using NDVI approach to validate this model is not enough. In addition, authors already mention that the study area has been used under intensive agriculture so, authors should have also given field experiment data or questionary-survey study or statistical real data by comparing between crop yield and SQI to make check or validate them.
Response: The vegetation cover of the study area is changing during the winter and spring seasons as the rainfall is active, whereas the natural vegetation spreads throughout the study area according to Mohamed et al. [17] and Mohamed et al. [28,61] and this confirmed by the field work. We improved NDVI map in Supplemental martial (Figure S2. NDVI of investigated area) to show that the cultivated area located in the small area of wadis, and it is logic according to the mentioned studies, so we depended on it to confirm our work and calculate SQ. Moreover, the results are correlated strongly with land capability classes of study area in the agreement of [2]. The reason of choose of study area is the regional government try to find new areas for cultivation to achieve food security.
---------------------------------------------------------------------------------------------------------------------
As for your conclusion, the results and discussion are insufficient, please describe more about the research results and it's better to make a series of important policy suggestions for policy makers.
Response: Some details were added in results and discussion section. We add the following paprgraph to conclusion section as suggestions to decision makers to improve SQ in the study area:
Improving SQ in the study area requires some agriculture practices for instance; reduce risks of soil erosion which might occurs due to, geomorphologic properties of study area, increase the rain water and fertilizers efficiency used as, the cultivation in the study area depends on winter rains and minimize losing of organic matter and nutrients.
---------------------------------------------------------------------------------------------------------------------
In addition, you should discuss any weaknesses or points that you have not considered in your paper.
Response: We added the following phrase to the conclustion section to consider it in the future: It is suggested that increasing field work and approaches of soil quality calculation in the future studies.
---------------------------------------------------------------------------------------------------------------------
would appreciate if the revised version of our manuscript would be considered for publication in Sustainability.
Sincerely,
Ebrahem M. Eid
Kafrelsheikh University, Kafr El-Sheikh, Egypt

Reviewer 2 Report
Dear authors, your study aims to propose and test a new GIS-based approach to assess soil quality in cropland under arid climatic conditions. This topic is important and has become a relevant issue in those countries where human population is growing quickly like Egypt and other African countries. The goal of achieving sustainable agriculture and high crop yield is challenging, and thus, the use of accurate indices to assess and map soil quality is a necessity. However, both the study and manuscript include many weaknesses and needs an in-depth refinement.
- The title is cumbersome. I propose this alternative: "A GIS-based approach for the quantitative assessment of soil quality and sustainable agriculture".
- Authors have used the terms "model" and "modelling", and this is not correct. Authors have used a tool included in a GIS software to automate the process of raster calculations. However, this is not a model. We can only use the terms "model" and "simulate" when we replicate the events that have happened in the past or will happen in the future or are currently happening in the present. Authors are proposing a set of indices to combine different attributes, parameters and properties, but they are not simulating or modelling any process of soil quality. Therefore, your contribution is a new GIS-based approach, but it is not a new model. The correct use of terms is important in a scientific publication.
- Abstract. There is a lack of information with regard to the main land uses and management practices in the study area. This information should be added as it is relevant to evaluate and interpret soil quality indicators.
- Abstract. First line: replace "is consider" by "is considered". In general, the manuscript is written correctly, but the use of English should be improved in some paragraphs. Please, review carefully the readability of the text.
- Abstract - discussion. There is a lack of information regarding the actual contribution of this study on the basis of existing literature in Egyptian and worldwide sites.
- Abstract. Authors talk about "agricultural sustainability", but they do not explain if they are talking about environmental sustainability, economic sustainability or social sustainability, or any combination of them. This aspect should be clarified.
- Keywords must be rewritten after addressing all comments.
- Introduction. First paragraph. Apart from the devastating consequences of lasting droughts, heavy floods, heat waves, and plagues, other relevant causes of hungry in Africa are wars and political instability, and the exponential and unprecedented growth of human population in the last decades. These aspects should be mentioned in order to provide a realistic overview of the situation.
- Introduction. Second paragraph. Soil quality is also influenced by the presence and concentration of pollutants: heavy metals, organic and inorganic components of fertilizers, pesticides and herbicides. These molecules have been massively used in farmland in the last decades worldwide, and thus, any index that aims to assess soil quality in cropland should include these aspects.
- Introduction. Last paragraph. Replace "geomorphologic units" by "topographic units".
- Introduction. Last paragraph. The point number 4 is not an objective of the study; it is the expected contribution of the study to the scientific literature. Thus, it has to be written out of the list of goals.
- Sections 2.1., 2.2. and 2.3. What are the main land uses, crop types and tillage practices of the study area? This information have to be clearly presented in the manuscript.
- Section 2.1. Authors said that the annual rainfall depth ranges between 100 and 200 mm. This is arid and hyper-arid conditions, and thus, the term "semi-arid" has to be deleted. Annual rainfall under semi-arid conditions ranges between 250 and 500 mm. Most Mediterranean landscapes have semi-arid climatic conditions with annual rainfall depths ca. 400 mm. Please, be careful with the use of terms.
- Figure S2. The NDVI is a dynamic index that changes every day of the year. I miss more information about the map that you have included in the manuscript (e.g. date, crop types), and why you have chosen this map among all NDVI maps.
- Figure 1. I suggest the following changes: I) reduce the size of the map of Egypt; and II) add two or three pictures showing the most important crops of the study area. Take in mind that you are talking about sustainable agriculture, but you are not showing how soils and fields look.
- Section 2.2. Authors said "Simple filter was used to decrease errors and noise of vegetation". However, no information is included regarding the tools or methods. This aspect should be improved.
- Figure 2. This map is not relevant. It should be better if you prepare a combination of maps in the same figure: Overlay the hillshade map (at the bottom), the DEM (in the middle with transparency) and the main land uses (in the top, as a vectorial map showing the boundaries of each land use). The new map will provide useful information about the physiographic conditions of the study area.
- Figure S4 is important and has to be included in the main text.
- The geomorphology of a landscape is the result of many processes and parameters, including lithology, climate and the long-term effect of overland and sub-surface flow processes. Therefore, it is not correct to use the GI "geomorphologic index" because you have only used topographic parameters, and thus, GI should be replaced by TI "topographic index".
- L.191. Replace "DVI" by "NDVI".
- The chemical index (CI). Again, I miss the addition of sub-factors associated with soil pollution. Authors have to consider that soil pollution is a common issue in farmland due to the abuse of chemicals.
- Figure 3. Why did you use interpolation to generate the topographic maps like the raster plan curvature, profile curvature, aspect, slope, etc.? These maps are directly generated from the corrected DEM. It is not necessary to use any interpolation technique. This workflow chart should be improved or explained better.
- Figure 4. Have you generated this map for this study? Is this map an actual contribution? Was this map included in a published document? This map includes more information than the "geomorphologic index".
- General comment. One indicator of soil quality is crop yield. In the same site, different fields with the same crop and managed following the same tillage practices, differences in productivity may be explained by changes in soil quality. However, authors have not included this aspect in any section of the manuscript.
- The number of references, 78 in total, is too high. I encourage authors to limit the number to 50 relevant studies.
Author Response
We appreciate the positive and constructive comments and suggestions provided by the reviewer on our manuscript. We have carefully studied the reviewer’ comments and have made revisions that are tracked in the revised version of the manuscript. We have tried our best to revise our manuscript according to the reviewer’ comments. Please find attached the revised version of our manuscript, which we would like to submit for your kind consideration. Once again, we would like to express our great appreciation to you and the reviewer for the comments on our manuscript.
Please find below our detailed responses to each of the points raised:
---------------------------------------------------------------------------------------------------------------------
Comments of Reviewer # 2:
The title is cumbersome. I propose this alternative: "A GIS-based approach for the quantitative assessment of soil quality and sustainable agriculture".
Response: Thanks so much Sir for your time and for positive comment. The title was changed accordingly.
---------------------------------------------------------------------------------------------------------------------
- Authors have used the terms "model" and "modelling", and this is not correct. Authors have used a tool included in a GIS software to automate the process of raster calculations. However, this is not a model. We can only use the terms "model" and "simulate" when we replicate the events that have happened in the past or will happen in the future or are currently happening in the present. Authors are proposing a set of indices to combine different attributes, parameters and properties, but they are not simulating or modelling any process of soil quality. Therefore, your contribution is a new GIS-based approach, but it is not a new model. The correct use of terms is important in a scientific publication- Abstract. There is a lack of information with regard to the main land uses and management practices in the study area. This information should be added as it is relevant to evaluate and interpret soil quality indicators.
Response: We feed the soil quality equations in GIS model builder (explained it in section 2,7) and this model is able provide useful tool for monitoring soil quality in the study under current condition thus we used the term of model, and it is commonly use d in previous studies such as
- Abuzaid, A.S.; Abdellatif, D.; Fadl, M. Modeling soil quality in Dakahlia Governorate, Egypt using GIS techniques. The Egyptian Journal of Remote Sensing and Space Sciences 2021, 24, 255-264.
Shokr, M.S.; Abdellatif, M.A.; El Baroudy, A.A.; Elnashar, A.; Ali, E.F.; Belal, A.A.; Attia, W.; Ahmed, M.; Aldosari, A.A.; Szantoi, Z.; et al. Development of a spatial model for soil quality assessment under arid and semi-arid conditions. Sustainability 2021, 13, 2893.
---------------------------------------------------------------------------------------------------------------------
- Abstract. First line: replace "is consider" by "is considered". In general, the manuscript is written correctly, but the use of English should be improved in some paragraphs. Please, review carefully the readability of the text.
Response: Many sections of paper were replaced and improved as recommended.
---------------------------------------------------------------------------------------------------------------------
- Abstract - discussion. There is a lack of information regarding the actual contribution of this study on the basis of existing literature in Egyptian and worldwide sites.
Response: We mentioned the following paragraph to the abstract:
…..it provides a full overview of SQ in the study area and can easily implement in the similar environment aimed at identifying soil quality challenges and fight the negative factors influence SQ in addition, achieving environmental sustainability, so this model can be applied on soils of countries that located in arid zone and have similar conditions…
---------------------------------------------------------------------------------------------------------------------
- Abstract. Authors talk about "agricultural sustainability", but they do not explain if they are talking about environmental sustainability, economic sustainability or social sustainability, or any combination of them. This aspect should be clarified.
Response: We added the following word environmental to clarify it.
---------------------------------------------------------------------------------------------------------------------
- Keywords must be rewritten after addressing all comments.
Response: We added some key words and improved them.
---------------------------------------------------------------------------------------------------------------------
- Introduction. First paragraph. Apart from the devastating consequences of lasting droughts, heavy floods, heat waves, and plagues, other relevant causes of hungry in Africa are wars and political instability, and the exponential and unprecedented growth of human population in the last decades. These aspects should be mentioned in order to provide a realistic overview of the situation.
Response: These aspects were added as recommended.
---------------------------------------------------------------------------------------------------------------------
- Introduction. Second paragraph. Soil quality is also influenced by the presence and concentration of pollutants: heavy metals, organic and inorganic components of fertilizers, pesticides and herbicides. These molecules have been massively used in farmland in the last decades worldwide, and thus, any index that aims to assess soil quality in cropland should include these aspects.
Response: These aspects were added as recommended.
---------------------------------------------------------------------------------------------------------------------
- Introduction. Last paragraph. Replace "geomorphologic units" by "topographic units".
Response: One of the purposes of this paper is to identify geomorphologic units and we already produced it in Figure 5 and areas of units in Table 2.
---------------------------------------------------------------------------------------------------------------------
- Introduction. Last paragraph. The point number 4 is not an objective of the study; it is the expected contribution of the study to the scientific literature. Thus, it has to be written out of the list of goals.
Response: The point number 4 was removed.
---------------------------------------------------------------------------------------------------------------------
- Sections 2.1., 2.2. and 2.3. What are the main land uses, crop types and tillage practices of the study area? This information have to be clearly presented in the manuscript.
Response: This information was given: ……this study is depending on winter rains and main crop is barely in some scattered fruits tree such as olive.
---------------------------------------------------------------------------------------------------------------------
- Section 2.1. Authors said that the annual rainfall depth ranges between 100 and 200 mm. This is arid and hyper-arid conditions, and thus, the term "semi-arid" has to be deleted. Annual rainfall under semi-arid conditions ranges between 250 and 500 mm. Most Mediterranean landscapes have semi-arid climatic conditions with annual rainfall depths ca. 400 mm. Please, be careful with the use of terms.
Response: The terms were used carefully. Thanks Sir.
---------------------------------------------------------------------------------------------------------------------
- Figure S2. The NDVI is a dynamic index that changes every day of the year. I miss more information about the map that you have included in the manuscript (e.g. date, crop types), and why you have chosen this map among all NDVI maps.
Response: We added the date, acquired in April 2020 as this study is rain fed area and this time of the year that reflect the vegetation cover of study area.
---------------------------------------------------------------------------------------------------------------------
- Figure 1. I suggest the following changes: I) reduce the size of the map of Egypt; and II) add two or three pictures showing the most important crops of the study area. Take in mind that you are talking about sustainable agriculture, but you are not showing how soils and fields look.
Response: All these changes were applied as recommended
---------------------------------------------------------------------------------------------------------------------
- Section 2.2. Authors said "Simple filter was used to decrease errors and noise of vegetation". However, no information is included regarding the tools or methods. This aspect should be improved.
Response: This aspect was improved as recommended please see it:
Simple filter by focal neighborhood statistics was used to decrease errors and noises. These noises occurred due to reclassification of topographic parameters. For the majority and mean focal statistic values, the most and average values of the specified neighborhood pixels were assigned to the canter pixel of the moving window
---------------------------------------------------------------------------------------------------------------------
- Figure 2. This map is not relevant. It should be better if you prepare a combination of maps in the same figure: Overlay the hillshade map (at the bottom), the DEM (in the middle with transparency) and the main land uses (in the top, as a vectorial map showing the boundaries of each land use). The new map will provide useful information about the physiographic conditions of the study area.
Response: We prepared a combination of maps as recommended.
---------------------------------------------------------------------------------------------------------------------
- Figure S4 is important and has to be included in the main text.
Response: We included Figure S4 in the text as recommended.
---------------------------------------------------------------------------------------------------------------------
- The geomorphology of a landscape is the result of many processes and parameters, including lithology, climate and the long-term effect of overland and sub-surface flow processes. Therefore, it is not correct to use the GI "geomorphologic index" because you have only used topographic parameters, and thus, GI should be replaced by TI "topographic index".
Response: The GI including the same parameters was used for spatial assessment of land sensitivity to degradation across Romania thus, for Scientific ethics we used the same concept, and we mentioned the reference in line 253.
Reference
Prăvălie, R.; Patriche, C.; Săvulescu, I.; Igor, S.; Georgeta, B.; Lucian, S. Spatial assessment of land sensitivity to degradation across Romania. A quantitative approach based on the modified MEDALUS methodology. Catena 2020,187, 104 - 407.
---------------------------------------------------------------------------------------------------------------------
- L.191. Replace "DVI" by "NDVI".
Response: DVI was corrected as recommended.
---------------------------------------------------------------------------------------------------------------------
- The chemical index (CI). Again, I miss the addition of sub-factors associated with soil pollution. Authors have to consider that soil pollution is a common issue in farmland due to the abuse of chemicals.
Response: This area is rained area and consider an important part of the existent economic activities in Egypt. Still there is no heavy cultivation in the study area so there is no need for using large amounts of pesticides or herbicides and there is not local factories in the study area which consider one of the major sources of soil contamination so we are not use this index in our study but we will keep it in our mind in the future studies after increasing of cultivation areas in the study area.
---------------------------------------------------------------------------------------------------------------------
- Figure 3. Why did you use interpolation to generate the topographic maps like the raster plan curvature, profile curvature, aspect, slope, etc.? These maps are directly generated from the corrected DEM. It is not necessary to use any interpolation technique. This workflow chart should be improved or explained better.
Response: Many thanks, we improved that workflow chart as recommended.
---------------------------------------------------------------------------------------------------------------------
- Figure 4. Have you generated this map for this study? Is this map an actual contribution? Was this map included in a published document? This map includes more information than the "geomorphologic index".
Response: This map is generated in this study and used for identifying geomorphologic units of study area and distribute soil profiles were based on these units which have different soil properties to try to reflect classes of soil quality in the study area as accurate as possible and the factors which used to calculate of GI also used to generate this map.
---------------------------------------------------------------------------------------------------------------------
- General comment. One indicator of soil quality is crop yield. In the same site, different fields with the same crop and managed following the same tillage practices, differences in productivity may be explained by changes in soil quality. However, authors have not included this aspect in any section of the manuscript.
Response: This area is rain fed area. The vegetation cover of the study area is changing during the winter and spring seasons as the rainfall is active, whereas the natural vegetation spreads throughout the study area according [30,32]and this confirmed by field work. We improved NDVI map in Supplemental martial (Figure S2. NDVI of investigated area) to show that the cultivated area located in the small area of wadis in addition land use and land cover (Figure 2) and it is logic according to the mentioned studies so we depended on it to confirm our work and calculate SQ moreover, the results are correlated strongly with land capability classes of study area in the agreement of [2] so we are not used the crop yield in this study.
---------------------------------------------------------------------------------------------------------------------
- The number of references, 78 in total, is too high. I encourage authors to limit the number to 50 relevant studies
Response: We developed model for assessment of soil quality, so we used a lot of references to confirm our work results moreover, other reviewers suggest improving introduction and discussion sections, so we added a lot of references.
---------------------------------------------------------------------------------------------------------------------
I would appreciate if the revised version of our manuscript would be considered for publication in Sustainability.
Sincerely,
Ebrahem M. Eid
Kafrelsheikh University, Kafr El-Sheikh, Egypt
Reviewer 3 Report
Evaluation for “sustainability-1454395”
A Developed Approach to the Quantitative Soil Quality Assessment for Potential Agricultural Sustainability based on GIS Modelling
This study tried to quantify the soil quality by using indicator approaches. It is a good initiative for precision agriculture. However, several unclear explanations in the Abstract, Introduction and method section need to be addressed before the next steps. Please see the comments below for the details.
The abstract would be better to extend, and the Materials and methods in Lines 25-28 need to be explained precisely. We know that there are several indexing methods and indicator and scoring methods. Thus they need to be clarified. The abstract should stand alone and be informative.
L29: how many soil quality classes are available? Why did SQ not start from “1”?
L30-31: the total area in these lines that were classified contains 94.22% of the study area? How about the rest of the study area? Were classified as miscellaneous?
L31: “The developed model shows” which model? We still did not see any information about the model type, structure, etc.! please clarify it.
L32: “kappa coefficient.” The evaluation method did not explain.
Keywords: please use diverse keywords that were not used in the title or the abstract to increase the manuscript's visibility.
L53-56 is not only defined by these limiting factors, and please see below articles in which diverse soil properties were used as quality indicators. In fact, based on the study purpose and the land use suitability and land use potential, the wide range of soil properties used in previous studies. Therefore, the input cannot be limited to a fixed set of variables.
Assessing the effects of deforestation and intensive agriculture on the soil quality through digital soil mapping. https://doi.org/10.1016/j.geoderma.2019.114139
Land use change effects on soil quality and biological fertility: A case study in northern Iran. https://doi.org/10.1016/j.ejsobi.2019.103119
So, I suggest the authors revise this section and include more soil properties.
L57: “Indices consider” please rewrite this sentence. “Indices” is vague.
L73 and L75: “GIS software” use the complete for when the first time used. “Geographic Information System (GIS)”
L84-91: the research gap did not clearly explain, and the novelty of the work was not explicit.
L109: “Normalized Difference Vegetation Index (NDVI)” how does this process? From which satellite source, Landsat, Sentinel? And please remove low and high.
L122: add a dot “.” after “GIS software [29]”
L132-133: why the profile description done by FAO and then classified by USDA? Though the USDA has the description for soil classification and on the field description manual!!
L138-139: the nutrients can be classified as chemical soil properties!
L165: move this section “2.5. Evaluation of geostatistical analysis” to the end of the M&M section.
L169: why DSQ? Do you mean ASQ?
L177: is there a reference for this model? Please add it.
L191: NDVI
equation (13): CaSO4 and CaCO3 did not report in the chemical properties in the measured materials and methods.
L207: Please move this section after section “2.2. Extracting landforms units”
L217: “Output of ach” what does it mean?
L220: “(i.e., very high, high, moderate and very low)” different levels reported in “L29-30” and Line398.
L365-367: This seems an incomplete sentence. Please revise it.
L371-372: the summation is not equal to 100%!
L395: chemical, biological, and …………..
Author Response
- November 2021
Anastasija Milenkovic
Assistant Editor
Sustainability
Dear Anastasija Milenkovic,
Please find attached the revised manuscript titled ‘A Developed Approach to the Quantitative Soil Quality Assessment for Potential Agricultural Sustainability Based on GIS Modelling’. Ms. Ref. No.: sustainability-1454395, authored by Abdellatif, M.A.; El Baroudy, A.A.; Arshad, M.; Mahmoud, E.K.; Saleh, A.M.; Moghanm, F.S.; Shaltout, K.H.; Eid, E.M.; and Shokr, M.S.
On behalf of my co-authors, I thank you very much for giving us the opportunity to revise our manuscript. We appreciate the positive and constructive comments and suggestions provided by the reviewers on our manuscript. We have carefully studied the reviewers’ comments and have made revisions that are tracked in the revised version of the manuscript. We have tried our best to revise our manuscript according to the reviewers’ comments. Please find attached the revised version of our manuscript, which we would like to submit for your kind consideration. Once again, we would like to express our great appreciation to you and the reviewers for the comments on our manuscript.
Please find below our detailed responses to each of the points raised:
---------------------------------------------------------------------------------------------------------------------
Comments of Reviewer # 2:
A Developed Approach to the Quantitative Soil Quality Assessment for Potential Agricultural Sustainability based on GIS Modelling
This study tried to quantify the soil quality by using indicator approaches. It is a good initiative for precision agriculture. However, several unclear explanations in the Abstract, Introduction and method section need to be addressed before the next steps. Please see the comments below for the details.
The abstract would be better to extend, and the Materials and methods in Lines 25-28 need to be explained precisely. We know that there are several indexing methods and indicator and scoring methods. Thus they need to be clarified. The abstract should stand alone and be informative.
Response: Thanks so much Sir for your time and for positive and constructive comments and suggestions. We addressed all of them as the following:
---------------------------------------------------------------------------------------------------------------------
L29: how many soil quality classes are available? Why did SQ not start from “1”?
Response: All classes of developed model (DSQ) are five classes i.e, very high (SQ1), high (SQ2), moderate (SQ3), low (SQ4) and very low (SQ5) while; study area located under four classes from SQ2 to SQ5.
---------------------------------------------------------------------------------------------------------------------
L30-31: the total area in these lines that were classified contains 94.22% of the study area? How about the rest of the study area? Were classified as miscellaneous?
Response: Because around 6% of study area is references terms i.e., table land, waterlogging and rock outcrop. These areas are not cultivated, and we added this information to the discussion.
---------------------------------------------------------------------------------------------------------------------
L31: “The developed model shows” which model? We still did not see any information about the model type, structure, etc.! please clarify it.
Response: Developed model is (DSQ) and structure of it explained in section 2.7 of martials.
---------------------------------------------------------------------------------------------------------------------
L32: “kappa coefficient.” The evaluation method did not explain.
Response: The kappa statics was used and explained in materials and methods section 2.8.
---------------------------------------------------------------------------------------------------------------------
Keywords: please use diverse keywords that were not used in the title or the abstract to increase the manuscript's visibility.
Response: We used some keywords that were not used in the title i.e., NDVI; GI.
---------------------------------------------------------------------------------------------------------------------
L53-56 is not only defined by these limiting factors, and please see below articles in which diverse soil properties were used as quality indicators. In fact, based on the study purpose and the land use suitability and land use potential, the wide range of soil properties used in previous studies. Therefore, the input cannot be limited to a fixed set of variables.
Assessing the effects of deforestation and intensive agriculture on the soil quality through digital soil mapping. https://doi.org/10.1016/j.geoderma.2019.114139
Land use change effects on soil quality and biological fertility: A case study in northern Iran. https://doi.org/10.1016/j.ejsobi.2019.103119
So, I suggest the authors revise this section and include more soil properties.
Response:
Response: Thanks Sir, we revised that section and added some soil properties as recommended.
---------------------------------------------------------------------------------------------------------------------
L57: “Indices consider” please rewrite this sentence. “Indices” is vague.
Response: This sentence was rewritten.
---------------------------------------------------------------------------------------------------------------------
L73 and L75: “GIS software” use the complete for when the first time used. “Geographic Information System (GIS)”
Response: It was corrected accordingly.
---------------------------------------------------------------------------------------------------------------------
L84-91: the research gap did not clearly explain, and the novelty of the work was not explicit.
Response: We added some details at the end of introduction to explain of the research gap and novelty of the work as the following:
The study area suffers from lack awareness of agriculture practices and scarcity of water as it depends on seasonal rain for cultivation [28]. Thus, the current study aims to: 1) Identify geomorphologic units of study area; 2) Quantitative assessment of soil quality using developed model based on four indicators i.e., chemical, physical, fertility and geomorphologic indices. The GI in addition, Normalized Difference Vegetation Index (NDVI) parameter were proposed and added as new factors in order to reflect the specific soil quality and categorize it into different classes as accurately as possible; 3) The results from this model validated with weight additive index and correlated with land capability. To our knowledge, very few studies that assess soil quality in the study area, so this paper will deliver valuable data to decision makers and regional governments to find the best ways for increasing soil quality and, overcome the food security problem which consider one of the most challenge for the 2030 Agenda for sustainable development.
---------------------------------------------------------------------------------------------------------------------
L109: “Normalized Difference Vegetation Index (NDVI)” how does this process? From which satellite source, Landsat, Sentinel? And please remove low and high.
Response: NDVI was calculated from Sentinel2 image and some details were added; please find it in methodology section under the title of Calculation of NDVI. Low and high has been removed from the map and classified it into six classes Figure S2.
---------------------------------------------------------------------------------------------------------------------
L122: add a dot “.” after “GIS software [29]”
Response: It was added as recommended.
---------------------------------------------------------------------------------------------------------------------
L132-133: why the profile description done by FAO and then classified by USDA? Though the USDA has the description for soil classification and on the field description manual!!
Response: It is very common in our soil studies of Egypt please find the below papers:
- Shokr, M.S.; Abdellatif, M.A.; El Baroudy, A.A.; Elnashar, A.; Ali, E.F.; Belal, A.A.; Attia, W.; Ahmed, M.; Aldosari, A.A.; Szantoi, Z.; et al. Development of a spatial model for soil quality assessment under arid and semi-arid conditions. Sustainability 2021, 13, 2893.
- El Baroudy, A.A. Mapping and evaluating land suitability using a GIS-based model. CATENA 2016,140, 96–104.
- Baroudy, A.A.E.; Ali, A.M.; Mohamed, E.S.; Moghanm, F.S.; Shokr, M.S.; Savin, I.; Poddubsky, A.; Ding, Z.; Kheir, A.M.S.; Aldosari, A.A.; Elfadaly, A.; Dokukin, P.; Lasaponara, R. Modeling Land Suitability for Rice Crop Using Remote Sensing and Soil Quality Indicators: The Case Study of the Nile Delta. Sustainability2020, 12, 9653. https://doi.org/10.3390/su12229653
- Said, M.E.S.; Ali, A.M.; Borin, M.; Abd-Elmabod, S.K.; Aldosari, A.A.; Khalil, M.M.N.; Abdel-Fattah, M.K. On the Use of Multivariate Analysis and Land Evaluation for Potential Agricultural Development of the Northwestern Coast of Egypt. Agronomy 2020, 10, 1318.
and we will consider it in the future publication
---------------------------------------------------------------------------------------------------------------------
L138-139: the nutrients can be classified as chemical soil properties!
Response: They are classified as biological or fertility properties according to:
- Shokr, M.S.; Abdellatif, M.A.; El Baroudy, A.A.; Elnashar, A.; Ali, E.F.; Belal, A.A.; Attia, W.; Ahmed, M.; Aldosari, A.A.; Szantoi, Z.; et al. Development of a spatial model for soil quality assessment under arid and semi-arid conditions. Sustainability 2021, 13, 2893.
- El Baroudy, A.A. Mapping and evaluating land suitability using a GIS-based model. CATENA 2016,140, 96–104.
- Bakhshandeh, E; Hossieni, M; Zeraatpisheh, M; Francaviglia, R. Land use change effects on soil quality and biological fertility: A case study in northern Iran, European Journal of Soil Biology, 2019, 95.
---------------------------------------------------------------------------------------------------------------------
L165: move this section “2.5. Evaluation of geostatistical analysis” to the end of the M&M section.
Response: It was moved.
---------------------------------------------------------------------------------------------------------------------
L169: why DSQ? Do you mean ASQ?
Response: Sorry for this mistake; yes we mean ASQ and it was corrected in the text.
---------------------------------------------------------------------------------------------------------------------
L177: is there a reference for this model? Please add it.
Response: We have been already added it please see:
[48] Prăvălie, R.; Patriche, C.; Săvulescu, I.; Igor, S.; Georgeta, B.; Lucian, S. Spatial assessment of land sensitivity to degradation across Romania. A quantitative approach based on the modified MEDALUS methodology. Catena 2020,187, 104 - 407.
---------------------------------------------------------------------------------------------------------------------
L191: NDVI
Response: Many thanks, we corrected it.
---------------------------------------------------------------------------------------------------------------------
equation (13): CaSO4 and CaCO3 did not report in the chemical properties in the measured materials and methods.
Response: Thank you for your notice, we reported them.
---------------------------------------------------------------------------------------------------------------------
L207: Please move this section after section “2.2. Extracting landforms units”
Response: We moved it.
---------------------------------------------------------------------------------------------------------------------
L217: “Output of ach” what does it mean?
Response: We corrected it to the output of each.
---------------------------------------------------------------------------------------------------------------------
L220: “(i.e., very high, high, moderate and very low)” different levels reported in “L29-30” and Line398.
Response: Many thanks for valuable comment; we corrected it to be as the following: reclassified into five classes (i.e., very high, high, moderate low and very low).
---------------------------------------------------------------------------------------------------------------------
L365-367: This seems an incomplete sentence. Please revise it.
Response: This sentence was completed.
---------------------------------------------------------------------------------------------------------------------
L371-372: the summation is not equal to 100%!
Response: Thank you for your valuable comment; Because around 6% of study area is references terms i.e., table land, waterlogging and rock outcrop. These areas are not cultivated and we explained it in the text.
---------------------------------------------------------------------------------------------------------------------
L395: chemical, biological, and …………..
Response: We did it, thanks.
---------------------------------------------------------------------------------------------------------------------
would appreciate if the revised version of our manuscript would be considered for publication in Sustainability.
Sincerely,
Ebrahem M. Eid
Kafrelsheikh University, Kafr El-Sheikh, Egypt

Reviewer 4 Report
The article is very basic and standard. A revised version can be successfully considered for publication.
I would suggest to
1) clarify the conceptual linkage with sustainability, I see it, but it is too implicit in the paper
2) increase the details about the construction of maps, including the geo-statistical algorithm adopted for regionalization. I suggest it is the IDW, considering the spatial pattern in most of these maps. Please give details, alternatives and better specification.
3) The work is truly descriptive, there is no reason to publish all these maps in the text, I would suggest to remove them to a specific appendix.
4) literature review is gross and not completely appropriate for this paper, and should be enriched substantially.
Thank you.
Author Response
- November 2021
Anastasija Milenkovic
Assistant Editor
Sustainability
Dear Anastasija Milenkovic,
Please find attached the revised manuscript titled ‘A Developed Approach to the Quantitative Soil Quality Assessment for Potential Agricultural Sustainability Based on GIS Modelling’. Ms. Ref. No.: sustainability-1454395, authored by Abdellatif, M.A.; El Baroudy, A.A.; Arshad, M.; Mahmoud, E.K.; Saleh, A.M.; Moghanm, F.S.; Shaltout, K.H.; Eid, E.M.; and Shokr, M.S.
On behalf of my co-authors, I thank you very much for giving us the opportunity to revise our manuscript. We appreciate the positive and constructive comments and suggestions provided by the reviewers on our manuscript. We have carefully studied the reviewers’ comments and have made revisions that are tracked in the revised version of the manuscript. We have tried our best to revise our manuscript according to the reviewers’ comments. Please find attached the revised version of our manuscript, which we would like to submit for your kind consideration. Once again, we would like to express our great appreciation to you and the reviewers for the comments on our manuscript.
Please find below our detailed responses to each of the points raised:
---------------------------------------------------------------------------------------------------------------------
Comments of Reviewer # 3:
The article is very basic and standard. A revised version can be successfully considered for publication.
I would suggest to:
- clarify the conceptual linkage with sustainability, I see it, but it is too implicit in the paper
Response: Thanks so much Sir for your time and for positive and constructive comments and suggestions. We clarified the conceptual linkage with sustainability by adding more details for instance; in the objects of paper and also in 3.4, 3.6, 3.7 sections.
---------------------------------------------------------------------------------------------------------------------
- increase the details about the construction of maps, including the geo-statistical algorithm adopted for regionalization. I suggest it is the IDW, considering the spatial pattern in most of these maps. Please give details, alternatives and better specification.
Response: Some details were added accordingly. It is kriging interpolation and the models which used in this study section 2.5 in materials and table 4 in results and discussion section.
---------------------------------------------------------------------------------------------------------------------
- The work is truly descriptive, there is no reason to publish all these maps in the text, I would suggest to remove them to a specific appendix.
Response: We transferred Figure 5 which includes 16 maps to supplementary file.
---------------------------------------------------------------------------------------------------------------------
- literature review is gross and not completely appropriate for this paper, and should be enriched substantially.
Response: Thank you very much for your comment; we tried to make it more specific and add some current impacted research.
---------------------------------------------------------------------------------------------------------------------
would appreciate if the revised version of our manuscript would be considered for publication in Sustainability.
Sincerely,
Ebrahem M. Eid
Kafrelsheikh University, Kafr El-Sheikh, Egypt

Round 2
Reviewer 1 Report
After reading detail in second version of manuscript, it can be seen that quality of manuscript was increased and also giving answers to referee’s questions were clear and logical in terms of scientific approach particularly about field validation or verification using crop yields. In addition authors added some important additional information into conclusion section.
Author Response
- November 2021
Anastasija Milenkovic
Assistant Editor
Sustainability
Dear Anastasija Milenkovic,
Please find attached the revised manuscript titled ‘A Developed Approach to the Quantitative Soil Quality Assessment for Potential Agricultural Sustainability Based on GIS Modelling’. Ms. Ref. No.: sustainability-1454395, authored by Abdellatif, M.A.; El Baroudy, A.A.; Arshad, M.; Mahmoud, E.K.; Saleh, A.M.; Moghanm, F.S.; Shaltout, K.H.; Eid, E.M.; and Shokr, M.S.
On behalf of my co-authors, I thank you very much for giving us the opportunity to revise our manuscript. We appreciate the positive and constructive comments and suggestions provided by the reviewers on our manuscript. We have carefully studied the reviewers’ comments and have made revisions that are tracked in the revised version of the manuscript. We have tried our best to revise our manuscript according to the reviewers’ comments. Please find attached the revised version of our manuscript, which we would like to submit for your kind consideration. Once again, we would like to express our great appreciation to you and the reviewers for the comments on our manuscript.
Please find below our detailed responses to each of the points raised:
---------------------------------------------------------------------------------------------------------------------
Comments of Reviewer # 1:
After reading detail in second version of manuscript, it can be seen that quality of manuscript was increased and also giving answers to referee’s questions were clear and logical in terms of scientific approach particularly about field validation or verification using crop yields. In addition authors added some important additional information into conclusion section.
Response: Thanks so much Sir for your time and for positive comment.
---------------------------------------------------------------------------------------------------------------------
I would appreciate if the revised version of our manuscript would be considered for publication in Sustainability.
Sincerely,
Ebrahem M. Eid
Kafrelsheikh University, Kafr El-Sheikh, Egypt

Reviewer 2 Report
Authors have addressed or argued to all my comments and suggestions.
Reviewer 3 Report
The authors addressed the previous comments accurately and the manuscript improved accordingly. However, still, there are some comments and correction which needs to be addressed before publication.
L38: the abbreviation of “DSQ” used for “developed spatial model” which is not sound good and also in Line 202 was used for “assessment of soil quality”. I suggest the authors thoroughly revise the manuscript for the abbreviations. Does “developed spatial model” mean “assessment of soil quality”?
L 44: "GI" in keywords is not appropriate since it is not common and well known. Please use the complete words.
L 70: "Indices" is not a method. Probably better to say “index indicators”.
L129: Please remove “in the agreement with Mohamed et al.” and keep the citation.
Figure 1: Please add a scale bar to the study area part.
Figure S4: Please change “Profiles Locations” to “Profile Locations”.
L314: “Soils properties” change to “Soil properties”.
Author Response
- November 2021
Anastasija Milenkovic
Assistant Editor
Sustainability
Dear Anastasija Milenkovic,
Please find attached the revised manuscript titled ‘A Developed Approach to the Quantitative Soil Quality Assessment for Potential Agricultural Sustainability Based on GIS Modelling’. Ms. Ref. No.: sustainability-1454395, authored by Abdellatif, M.A.; El Baroudy, A.A.; Arshad, M.; Mahmoud, E.K.; Saleh, A.M.; Moghanm, F.S.; Shaltout, K.H.; Eid, E.M.; and Shokr, M.S.
On behalf of my co-authors, I thank you very much for giving us the opportunity to revise our manuscript. We appreciate the positive and constructive comments and suggestions provided by the reviewers on our manuscript. We have carefully studied the reviewers’ comments and have made revisions that are tracked in the revised version of the manuscript. We have tried our best to revise our manuscript according to the reviewers’ comments. Please find attached the revised version of our manuscript, which we would like to submit for your kind consideration. Once again, we would like to express our great appreciation to you and the reviewers for the comments on our manuscript.
Please find below our detailed responses to each of the points raised:
---------------------------------------------------------------------------------------------------------------------
Comments of Reviewer # 2:
The authors addressed the previous comments accurately and the manuscript improved accordingly. However, still, there are some comments and correction which needs to be addressed before publication.
Response: Thanks so much Sir for your time and for positive and constructive comments and suggestions. We addressed all of them as the following:
---------------------------------------------------------------------------------------------------------------------
L38: the abbreviation of “DSQ” used for “developed spatial model” which is not sound good and also in Line 202 was used for “assessment of soil quality”. I suggest the authors thoroughly revise the manuscript for the abbreviations. Does “developed spatial model” mean “assessment of soil quality”?
Response: We revised the abbreviation and used the DSQM for developed soil quality model and removed ASQ.
---------------------------------------------------------------------------------------------------------------------
L 44: "GI" in keywords is not appropriate since it is not common and well known. Please use the complete words.
Response: We changed it to Geomorphologic index instead of GI.
---------------------------------------------------------------------------------------------------------------------
L 70: "Indices" is not a method. Probably better to say “index indicators”.
Response: It was changed as you recommended.
---------------------------------------------------------------------------------------------------------------------
L129: Please remove “in the agreement with Mohamed et al.” and keep the citation.
Response: It was removed.
---------------------------------------------------------------------------------------------------------------------
Figure 1: Please add a scale bar to the study area part.
Response: It was Added as recommended.
---------------------------------------------------------------------------------------------------------------------
Figure S4: Please change “Profiles Locations” to “Profile Locations”.
Response: It was changed as recommended.
---------------------------------------------------------------------------------------------------------------------
L314: “Soils properties” change to “Soil properties”.
Response: It was changed as recommended.
---------------------------------------------------------------------------------------------------------------------
I would appreciate if the revised version of our manuscript would be considered for publication in Sustainability.
Sincerely,
Ebrahem M. Eid
Kafrelsheikh University, Kafr El-Sheikh, Egypt

Reviewer 4 Report
Godd revision overall. Approved
Author Response
- November 2021
Anastasija Milenkovic
Assistant Editor
Sustainability
Dear Anastasija Milenkovic,
Please find attached the revised manuscript titled ‘A Developed Approach to the Quantitative Soil Quality Assessment for Potential Agricultural Sustainability Based on GIS Modelling’. Ms. Ref. No.: sustainability-1454395, authored by Abdellatif, M.A.; El Baroudy, A.A.; Arshad, M.; Mahmoud, E.K.; Saleh, A.M.; Moghanm, F.S.; Shaltout, K.H.; Eid, E.M.; and Shokr, M.S.
On behalf of my co-authors, I thank you very much for giving us the opportunity to revise our manuscript. We appreciate the positive and constructive comments and suggestions provided by the reviewers on our manuscript. We have carefully studied the reviewers’ comments and have made revisions that are tracked in the revised version of the manuscript. We have tried our best to revise our manuscript according to the reviewers’ comments. Please find attached the revised version of our manuscript, which we would like to submit for your kind consideration. Once again, we would like to express our great appreciation to you and the reviewers for the comments on our manuscript.
Please find below our detailed responses to each of the points raised:
---------------------------------------------------------------------------------------------------------------------
Comments of Reviewer # 3:
Good revision overall. Approved
Response: Thanks so much Sir for your time and for positive comment.
---------------------------------------------------------------------------------------------------------------------
I would appreciate if the revised version of our manuscript would be considered for publication in Sustainability.
Sincerely,
Ebrahem M. Eid
Kafrelsheikh University, Kafr El-Sheikh, Egypt
